# Pol θ-mediated end-joining uses microhomologies containing mismatches

Yuzhen Li ⏺, Ngoc K. Dang, Wei He ⏺, Mark Returan, Denisse Carvajal-Maldonado ⏺, Adele T. Guerin ⏺, Han Xu ⏺, Bin Liu ⏺ & Richard D. Wood ⏺ ✉

DNA polymerase theta (Pol θ) initiates repair of DNA double-strand breaks by pairing single strands at short "microhomologies". It is important to understand microhomology selection, as some cancer cells rely on Pol θ for survival. Here, we investigate end-joining by purified human Pol θ, employing DNA sequencing of products generated from oligonucleotide libraries having diverse 3′ ends. Pol θ overwhelmingly selects short internal microhomologies found within 15 nucleotides of the terminus of single-stranded DNAs, restricting deletion size during end-joining. Significantly, we find that the selected microhomologies are usually interrupted by mismatches and that base pairing within 6 nucleotides of the 3′ end is important for determining microhomology choice. Bidirectional synthesis is not necessary to initiate end-joining. The preference for mismatched microhomologies suggests a revision of the definition of microhomology to account for the unique properties of Pol θ. This could advance the analysis of mutations in cancer genomes.

DNA double-strand breaks (DSBs) can cause catastrophic damage to chromosomes if not repaired appropriately. One crucial pathway for DSB repair is DNA polymerase theta (Pol θ)-mediated end-joining (TMEJ). TMEJ maintains genomic stability during early development and helps defend against exposure to DNA-damaging agents[1,2]. It is also essential for repairing DSB when homologous recombination or non-homologous end-joining pathways are compromised[3].

In human cells, Pol θ is a 290 kDa protein with an N-terminal helicase-like domain connected to a C-terminal polymerase domain via a linker domain[1,2,4] (Fig. 1a). During TMEJ, Pol θ initiates end-joining between two single-stranded DNA tails[4–8] at a "microhomology" (MH), typically defined as 2–6 consecutively matched base pairs[2,4,9]. It is unlikely that a random break in the genome will have a perfectly matched MH at the break ends. Therefore, internal MHs located within the 3′ tails rather than at the termini are primarily used during TMEJ (Fig. 1b). Most in vitro TMEJ experiments have used a DNA polymerase domain fragment of Pol θ, and ssDNAs containing terminal matched MHs or long internal MH[7,10]. A study using short oligonucleotides (mostly purines or pyrimidines only) showed that the purified polymerase domain has a remarkable ability to prime using only a few bp[8]. However, there is no direct evidence, such as from DNA sequencing, to confirm that the designed microhomologies are used by Pol θ for end-joining.

There are important unanswered questions regarding the ability of Pol θ to initiate priming at internal microhomologies. There are suggestions that internal mismatches might be permitted at a site of microhomology[7,8], but it is not known if such microhomologies are frequently selected by Pol θ, and where mismatches are permitted near a terminus. Under some circumstances, Pol θ can extend from a mismatched 3′ terminus[11], but it is unclear whether this often occurs during TMEJ. The current standard concept defining MH for TMEJ of 2–6 consecutive bp MH is insufficient to account for some Pol θ-dependent repair events, which appear to lack MH, or only have 1 bp MH[12]. Further, we need to know if Pol θ normally extends both strands bidirectionally from a microhomology.

The aim of these studies was to help answer these questions by determining the microhomology selection preferences of purified human Pol θ, by itself, with defined DNA. We designed sets of oligonucleotides or oligonucleotide libraries with diverse 3′ ends to initiate end-joining and determined outcomes by analyzing data derived from DNA sequencing. Although simple in conception, the work reveals unique pairing and extension capabilities of Pol θ. Microhomologies

Department of Epigenetics and Molecular Carcinogenesis, MD Anderson Cancer Center, Houston, TX, USA. ✉e-mail: rwood@mdanderson.org

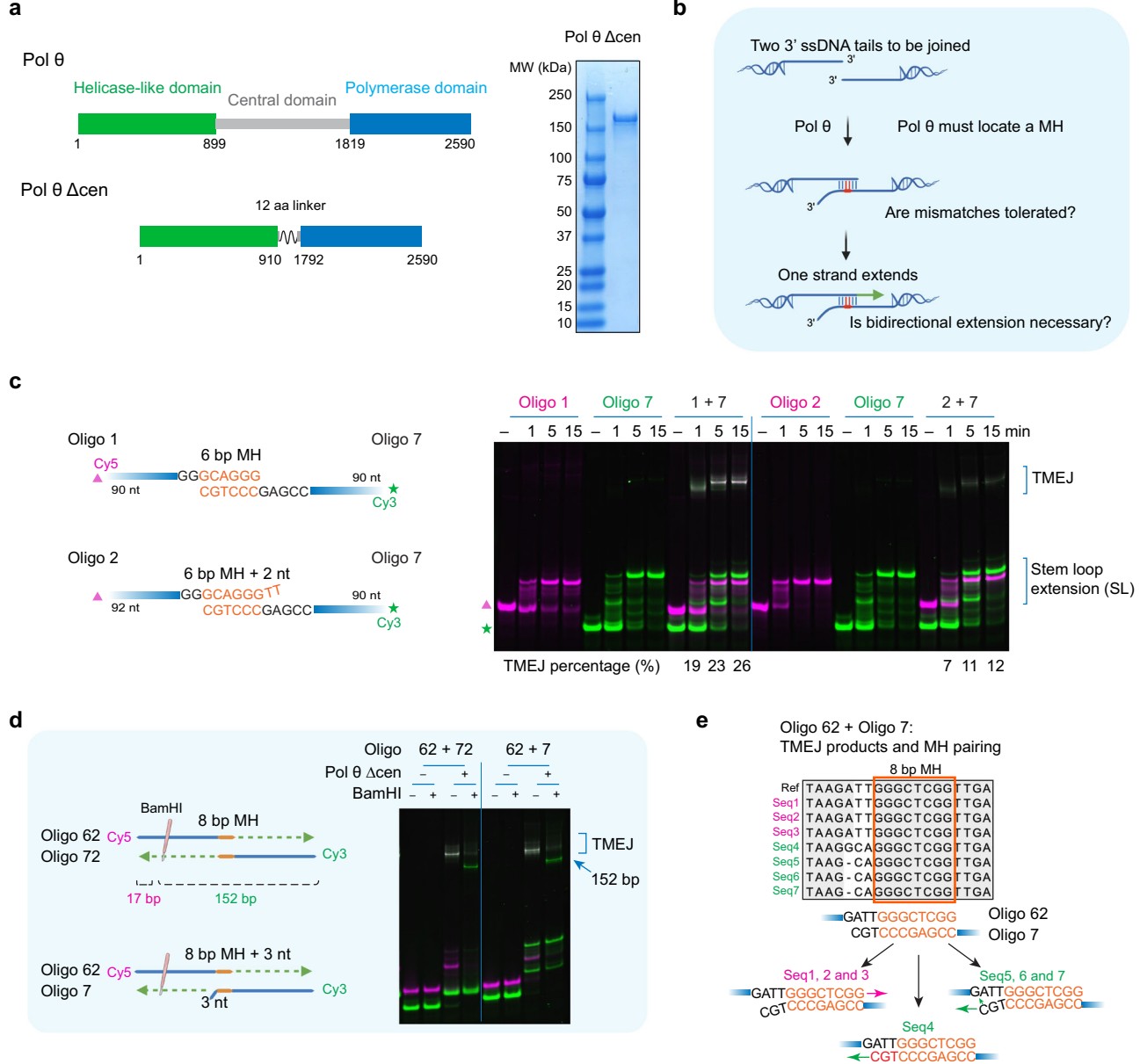

**Fig. 1 | Pol θ can perform end-joining with diverse paired ssDNAs. a** Schematic of Pol θ Δcen. Domains are shown with amino acid boundaries for human Pol θ. The polyacrylamide gel shows purified human Pol θ Δcen, which was similar in at least three different purification runs. **b** Overall TMEJ process and questions to be answered. Created in BioRender. Li, Y. (2025) https://BioRender.com/vxre3ij. **c** End-joining of two sets of paired oligonucleotides by Pol θ Δcen. The schematic shows two sets of paired oligonucleotides containing 5′ fluorescent labels and 3′ terminal or internal 6 bp MH. Reaction mixtures incubated at different time points were separated by electrophoresis on a native 10% polyacrylamide gel (*n* = 2). Intensities were measured with ImageJ, and the percentage of TMEJ products is labeled under the corresponding lanes. A magenta triangle indicates the starting Cy5-labeled oligonucleotide, and a green star indicates the starting Cy3-labeled oligonucleotide. **d** Verification of end-joining products from a single experiment. The schematic shows a diagnostic BamHI site and the fragment size of end-joining products after digestion. Created in BioRender. Li, Y. (2025) https://BioRender.com/vxre3ij. The fluorescence colors of the fragments are indicated. Electrophoresis of end-joining products was conducted with or without BamHI digestion. The digested large fragment (~152 bp) is labeled with an arrow. **e** Sequence alignment of products after end-joining of Oligo 62 and Oligo 7, which have three unpaired nucleotides adjacent to an 8 bp MH, as shown at the bottom. Source data are provided as a Source Data file.

containing terminal mismatches are usually not selected. Instead, internal MH located within 15 nucleotides of the ssDNA 3′ end are usually preferred. Bidirectional extension by Pol θ is not necessary, as extension of one strand is sufficient to initiate a successful TMEJ event. Most importantly, we show that Pol θ usually selects microhomologies that contain one or more mismatches. This has a major consequence as it suggests a redefinition of the concept of microhomology, for TMEJ. These results will enable a better understanding and recognition of Pol θ-mediated repair outcomes in normal and cancer cells.

## Results

### Pol θ extends from matched and mismatched oligonucleotides

We sought to understand the preference of Pol θ for selecting MH during TMEJ. To do this, we developed end-joining reactions using two ~90-mer oligonucleotides and purified human Pol θ Δcen (Fig. 1a), an active construct of Pol θ which contains the helicase-like and polymerase domains separated by a shorter linker to aid purification[13]. One oligonucleotide was 5′-labeled with fluorescent dye Cy5, and the other with Cy3 (Fig. 1c). After incubating the two oligonucleotides with Pol θ

Δcen in a reaction buffer for either 1, 5, and 15 min, reaction products were separated on a native 10% polyacrylamide gel. Products arising from extension by Pol θ are observed as a white band arising from the overlap of green and magenta signals (Fig. 1). Such products can be seen when oligonucleotides are used with a designed 4, 6, or 8 bp terminal MH (Fig. 1c–e and Supplementary Fig. 1).

Aside from TMEJ products, each long oligonucleotide also yields stem-loop extension products (extended green and magenta bands), which are also generated after incubation of the individual oligonucleotides with Pol θ[14]. Most stem-loop products are not accessible for further end-joining by Pol θ alone, as shown by combining reaction mixtures of ssDNA substrates separately preincubated with Pol θ (Supplementary Fig. 1b).

Joining also occurred (with varying efficiencies) when the oligonucleotides were mismatched at the 3′ terminus (Fig. 1c). The example in Fig. 1d shows the special case where the ssDNAs have an 8 bp microhomology near the terminus. TMEJ bands were observed when the oligonucleotides had a completely matched terminal microhomology, or when the "bottom" strand had three mismatched nucleotides at the 3′ end (Fig. 1d). In both cases, the bottom strand was fully extended, as shown by susceptibility of the TMEJ product band to cleavage by BamHI restriction endonuclease (Fig. 1d). The two strands were also fully extended in a substrate with a 4 bp terminal MH, because the TMEJ products were sensitive to cleavage on both sides by restriction enzymes (Supplementary Fig. 1c, d). DNA sequencing (Supplementary Fig. 2) was used to determine the products of end-joining for the reaction where the three 3′ terminal bases of the bottom strand were unpaired. Both strands were extended with similar efficiency, as three of the seven sequences matched extension from the top strand, and four from the bottom (Fig. 1e). The sequences extended from the bottom strand showed one sequence consistent with direct mismatch extension (Fig. 1e, seq 4), and three with a 1-nucleotide deletion that likely arose from loopout priming (Fig. 1e, seq 5–7). These results suggest that Pol θ can perform end joining on designed ssDNA substrates with either matched or mismatched bases.

## Internal MHs are selected by Pol θ in preference to a short core 4 bp MH

We investigated the outcomes of TMEJ when a shorter MH was provided, modeling the more commonly encountered situation in vivo. Figure 2a shows a set of two oligonucleotides which have a 4 bp MH, with two additional unpaired 3′ terminal bases on the top strand. The oligonucleotide nomenclature 2N2M indicates the two additional bases (2N), both mismatched (2M). On a native gel, multiple TMEJ products are observed (Fig. 2a). DNA sequencing showed that for both the top and bottom strands, Pol θ rarely extended from the 4 bp MH near the termini, but instead chose a new MH ("anchoring position") (Fig. 2a, b and Supplementary Fig. 8a). For top strand extension, a major outcome is an extension from an internal 9 bp MH interrupted by two mismatches (Fig. 2b). Of 16 sequences analyzed for extension of the bottom strand, 5 resulted by extension from the provided 4 bp MH. However, it was more common for Pol θ to select alternative internal anchoring positions with apparently better base pairing (Fig. 2b). When internal anchoring occurs during repair of a DSB, it causes short deletions, a hallmark of TMEJ[6,15].

We sought to set up an analysis by which top and bottom strand anchoring positions could be determined unambiguously. This was done by blocking the Cy3-labeled bottom strand extension with a 3′-phosphate group (Fig. 2c, Oligo 10), or by substituting three T residues with U in the bottom strand, facilitating subsequent enzymatic removal of the bottom strand before sequencing analysis (Fig. 2c, Oligo 9). After incubation with a Cy5-labeled oligonucleotide containing a complementary 4 bp terminal MH, Pol θ performed extension

to give end-joining products that migrate slowly on the gel, even when only one strand is extended. As polymerization is blocked by the 3′-phosphate, the slightly slower migration of TMEJ products with Oligo 10 compared to Oligo 7 (Fig. 2c) may arise from a secondary structure remaining in the single-stranded DNA of Oligo 10. This approach was checked further using an oligonucleotide pair having two unmatched bases at the 3′ end of the top strand (Fig. 2d). Sequencing showed that similar deletion patterns were obtained for the top strand when bottom strand polymerization was blocked by a 3′ phosphate or when the bottom strand was inactivated by uracil targeting (Fig. 2d and Supplementary Fig. 8b). These results indicate that Pol θ preferentially selects internal MH with better base pairing in preference to the designed 4 bp core MH, and that blocking extension on one strand does not impede Pol θ's use of the other strand.

## Imperfectly paired internal MHs are selected in preference to the extension of mismatched ends

To further understand how unpaired 3′-ends affect MH selection, a series of oligonucleotides was designed with a core 4 or 6 bp MH and mismatched ends on the top strand. With a 6 bp terminally matched MH, Pol θ readily performed end-joining, as did E. coli DNA polymerase I Klenow fragment (Kf exo-), a well-studied DNA polymerase from the same A-family[16] (Supplementary Fig. 3a, c). With one to four 3′ mismatches on the top strand following the 6 bp MH, Pol θ still performed end-joining, but at reduced efficiency (Supplementary Fig. 3b). When one end was mismatched, Kf exo- could not extend either strand and no end-joining products were produced.

With a 4 bp MH available, Pol θ also could produce TMEJ products visible on a native gel while Kf exo- could not (Fig. 3a, b and Supplementary Fig. 3h). The overall yield of joined products was similar when 1 or 2 terminal mismatches were added, but fewer products were formed when there were more than two terminal mismatched bases. The pattern of products on the gel was different in each case (Fig. 3b and Supplementary Fig. 3h).

DNA sequencing of products for the oligonucleotides with the 4 bp terminal MH (Fig. 3c), showed that the major products arise from the terminal MH. However, other TMEJ events frequently occur by alternative anchoring with the bottom strand. The favored MH in these cases had more matched bases but with an internal mismatched base (Fig. 3c and Supplementary Fig. 8c). When unpaired bases were appended to the 4 bp MH, new MHs containing internal mismatches were preferred for both strands as further anchoring positions became possible, as shown by the example with three unpaired bases in Fig. 3d and Supplementary Fig. 8d. The majority of selected MHs arise from the bottom strand. When additional unpaired bases were appended to the 4 bp core MH, almost all selected MHs were internal (Supplementary Fig. 3d, e). Similarly, for the 6 bp core MH with one to four unpaired bases, all selected MHs are also internal (Supplementary Fig. 3f, g). Therefore, Pol θ favors the use of imperfectly paired internal MH over extension from mismatched ends.

Pol θ has an ATPase activity associated with its helicase-like domain[17]. To assess the possible impact of this activity, we conducted experiments with or without ATP (Supplementary Fig. 9a, b). Sequencing analysis of TMEJ outcomes showed that ATP has no significant effect on terminal and internal MH selection (Supplementary Fig. 9c), or on the overall pattern of MH anchoring (Supplementary Fig. 9d–i). Pol θ was also unable to unwind stable stem-loop products in the presence of nucleotide triphosphates (Supplementary Fig. 9b), as shown earlier[14].

In vitro, TMEJ is more efficient with a greater number of paired bases and hydrogen bonds within an MH[10]. Our data (e.g., Fig. 3c, d) reveal an additional feature: MHs used by Pol θ frequently include an internal mismatch. This implies that many more MH opportunities are available than originally envisaged, because Pol θ often selects MH that contain mismatches.

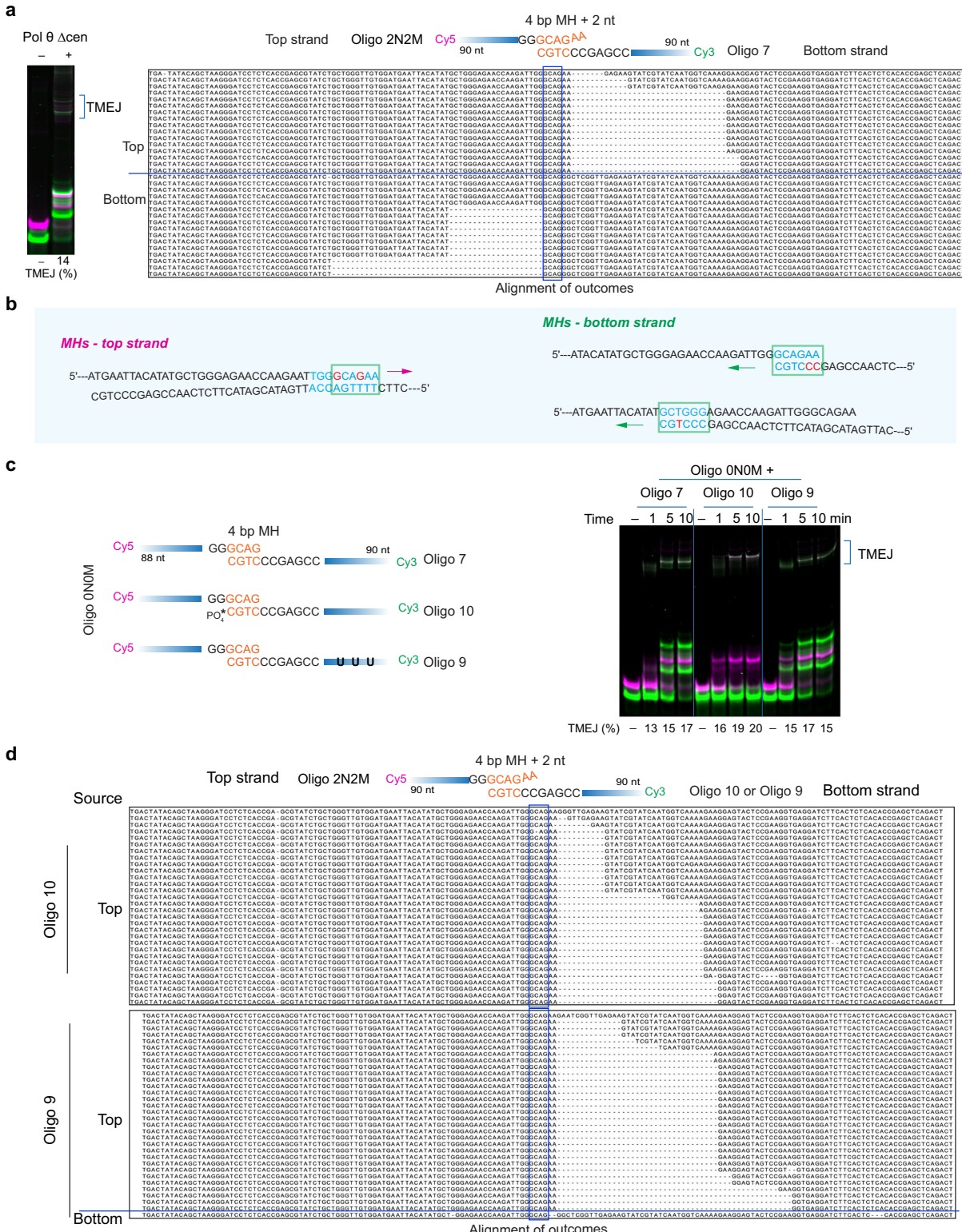

**MHs are usually selected by Pol θ within 15 nucleotides of the 3′ end**

To better understand how Pol θ selects MHs for end-joining, we analyzed large pools of oligonucleotides. Top strand oligonucleotide libraries were designed with a core 4 bp MH with random 3′ terminal ends, 0–5 nt in length (Fig. 4a). High-throughput sequencing allows identification of cases where the preexisting 4 bp microhomology is

used, and cases where internal microhomologies are chosen that make use of the additional 3′ bases. Each top strand oligonucleotide has an individual barcode, allowing subsequent identification of the extended strand.

A bottom strand was synthesized with either a normal 3′ end or containing a blocking 3′ phosphate group. By blocking bottom strand extension, it can be unambiguously determined that extension

**Fig. 2 | Pol θ selects internal MHs in preference to a shorter 4 bp core MH.**
**a** Electrophoresis of end-joining reactions with the schematic showing paired oli-
gonucleotides containing a 4 bp MH followed by two mismatches ($n = 2$). The
percentage of TMEJ products is labeled under the lanes. The alignment of TMEJ
outcomes from a single experiment is shown. The blue box indicates the 4 bp MH
and a blue line separates top strand TMEJ outcomes (top) and bottom strand TMEJ
outcomes (bottom). A larger typeface version of the alignment is in Supplementary
Fig. 8a. **b** Anchoring positions and local pairing information of major MHs (colored
brackets in panel a) used by Pol θ Δcen. The arrow indicates the extension direction
of Pol θ Δcen. A green square labels 6 bp of local MH, and blue-shaded bases show
an adjacent extended MH. Red nucleotides indicate mismatches. **c** Electrophoresis
of end-joining reactions by Pol θ Δcen of a top oligonucleotide (Oligo 0N0M) and
different bottom oligonucleotides from a single experiment. The schematic shows
paired oligonucleotides containing terminal 4 bp MH. Oligo 10 and 9 have the same

sequence as Oligo 7. Oligo 10 is modified at the 3′ end with a phosphate group
(*$PO_4$). U in Oligo 9 indicates 3 different T nucleotides replaced by U. The per-
centage of TMEJ products is labeled under the lanes. **d** End-joining outcomes
alignment of two different paired oligonucleotides (Oligo 2N2M and Oligo 10, Oligo
2N2M and Oligo 9) by Pol θ Δcen. The schematic shows paired oligonucleotides
containing an internal 4 bp MH with two mismatches. A blue box indicates the 4 bp
MH and a blue line distinguishes top strand TMEJ outcomes (top) and bottom
strand TMEJ outcomes (bottom). A larger typeface version of the alignment is in
Supplementary Fig. 8b. Before clean-up for amplification and sequencing, TMEJ
mixtures with Oligo 9 and Oligo 2N2M were treated with the USER enzyme, which
contains uracil-DNA glycosylase and the DNA glycosylase-lyase Endonuclease VIII,
to remove uracil in Oligo 9. TMEJ mixtures with Oligo 10 and Oligo 2N2M were
directly cleaned up for amplification and sequencing. Source data are provided as a
Source Data file.

products arose from the top strand. After an end-joining reaction with
Pol θ, the products were prepared for high-throughput sequencing
(Fig. 4b). The outcomes are analyzed in detail below, but first we note
the high correlation between overall outcomes of experiments with an
unblocked bottom strand (Oligo 8) and outcomes with a blocked
bottom strand (Oligo 10) (Fig. 4c). In experiments using oligonucleo-
tide pools for the top strand and a blocked bottom strand (Oligo 10)
the overall pattern and number of read counts arising from each oli-
gonucleotide pool were as expected from the library design (Supple-
mentary Fig. 4a). Reads were filtered for errors (defined in
Supplementary Methods) and then each remaining read was categor-
ized as a deletion, insertion, or a minor category of complex events
with mismatches to the reference sequence in the join sequence at
multiple positions.

The major end-joining outcomes were deletions, arising from
internal anchoring positions selected by Pol θ (Fig. 5a–c). At a lower
frequency, insertions and complex events also occur (Fig. 5a and
Supplementary Fig. 4b). Similar results were obtained when using the
unblocked bottom strand (Supplementary Fig. 5a, b). A small fraction
of end-joining outcomes were classified as cases where the 3′ end of
the extending oligonucleotide appears to have been shortened
(Supplementary Fig. 4c). These outcomes are not considered further
in the analysis, as it is likely that these arose from a low level of
incomplete oligonucleotide synthesis, or minor reagent contamina-
tion. Pol θ does not have intrinsic exonuclease activity[18], and nuclease
activity is not detected for the Pol θ protein used here (Supplemen-
tary Fig. 4d).

To characterize the deletions formed by Pol θ, the anchoring
position was formally defined as the distance (number of bases)
between the 3′ ends of the joining oligonucleotides (Fig. 5b). Each
position within the MH was also coded for later base pairing analysis. In
vivo, repair based on specific extra-chromosomal substrates or on
DNA double-strand breaks introduced at specific sites in a chromo-
some shows that TMEJ usually uses a MH within 15 nt of the two ssDNA
termini[5]. Our data with purified Pol θ and DNA are consistent with this
observation: the selected MHs are largely within 15 nucleotides of the
3′ end (Fig. 5c, d and Supplementary Fig. 5c). Even though the analysis
allowed detection of anchoring up to position 40, a higher percentage
(~80%) of MHs within 6–15 nt were selected by Pol θ as the diversity of
3′ ends increased (Fig. 5d).

**Selected MH usually contains mismatches**
The small-scale sequencing results (Figs. 2a, b, 3c, d) suggested that
~6 bp contribute to MH selection. This is consistent with the fact that
the DNA polymerase active site of Pol θ contains basic residues that
contact the six terminal phosphate backbone positions of the primer[19].

It is important to determine how base pair matching within this
region influences MH selection by Pol θ. To set up this analysis, we
defined bases in the anchoring position as P1–P6, with P1 as the
nucleotide at the 3′ end of the extended strand (Fig. 6a). Deletion

outcomes for high-throughput sequencing Libraries 6−10 were then
clustered into different types according to the number of matches
within the P1–P6 region (Fig. 6b). In the low-complexity Library 6 (only
5 oligonucleotides, Fig. 4a), the major end-joining products arose from
4 or 5 bp terminal MHs, consistent with Fig. 5c. With the higher com-
plexity Libraries 8-10, the analysis reveals that the MH anchoring
positions selected by Pol θ usually contain 2–5 matched bases (and
thus 1–4 mismatches) within P1–P6 (Fig. 6b).

Although each anchoring position usually contains multiple
matched base pairs, these matches are usually not consecutive. This
can be visualized by clustering the favored MH anchoring types for
each top strand oligonucleotide paired with Oligo 10 (Library 6–10,
Fig. 6c). Almost all the favored MHs were interrupted by mismatches
(Fig. 6c), where blue positions indicate matches and white positions
are mismatches. The same conclusion applies to the unblocked Oligo 8
(Library 1–5, Supplementary Fig. 6a). Small-scale sequencing also
showed that the favored microhomologies usually had interrupted
base pairing (Figs. 2b, 3c, d).

For a higher resolution analysis of the positions where base pair
matching is most important, we analyzed outcomes from Library 10.
This is the most diverse library, with 5 random nucleotides at the 3′
end of the top primer strand (1024 combinations). We compared the
outcomes to theoretical random anchoring using the 35
potential P1 sites on the bottom strand, Oligo 10. (The first 5 P1 sites
of the 3′ end were not analyzed as the anchored MH was shorter
than 6 bp). The fraction of total observed reads that were matched
at each potential P1–P5 site was calculated and compared to the
theoretical ratio. In a theoretical random distribution, each position
will have a match frequency of 0.25, and a mismatch frequency of
0.75. In the observed Pol θ selected MH, 0.93 of the outcomes are
matched at P1 (Fig. 6d). The same analysis was carried out for
positions P2– P5. Matching each other position also had an
observed frequency higher than theoretical (Supplementary Fig.
6b), with the highest probabilities towards the 3′ end (median
values: $P1 = 0.93$, $P2 = 0.83$, $P3 = 0.66$, $P4 = 0.55$, $P5 = 0.5$).

As P1 matching was most important for MH selection, we then
tested whether matched pairing at additional positions (P2–P5) is more
likely at anchoring positions when P1 is already matched. Figure 6e
shows all cases where P1 is matched, separating into categories where
there are 1, 2, 3, or 4 additional matches in the anchoring position
chosen by Pol θ. Significantly, outcomes with P1 matched and 2 or
more additional matches occurred at much higher than expected
frequencies. The fraction of outcomes where only P1 is matched is
much lower than expected. Therefore, Pol θ preferentially selects MH
with matches at P1 plus additional matches within P2–P5, but the
matches do not need to be consecutive.

To consider position P6, the analysis was adjusted because this
nucleotide is fixed as a G in the primer strand. We first considered the
cases where P6 is mismatched. If outcomes were random, P1 would be
paired in 0.25, and not paired in 0.75 (red dots, Supplementary Fig.

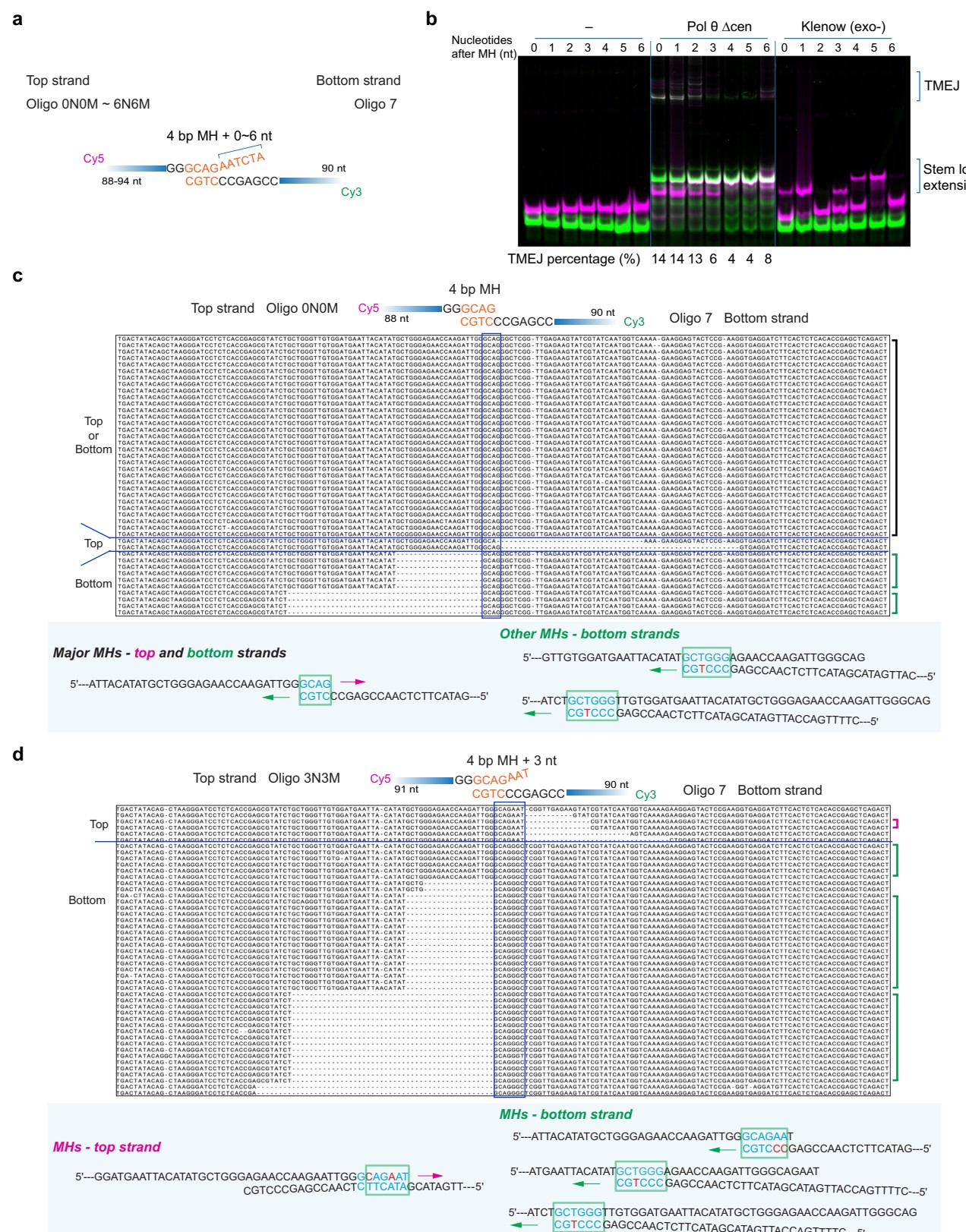

6c). The results for Pol θ-mediated experimental outcomes were similar to Fig. 6d: at position P1, 0.9 of the outcomes are matched. Additional matching at one or more positions also improved the chances of anchoring above random expectation (Supplementary Fig. 6d). For the cases where P6 is matched, P1 matching is also highly favored for successful anchoring (Supplementary Fig. 6e), and

additional matches improve the chances of anchoring (Supplementary Fig. 6f).

The analysis above shows that although mismatches can be tolerated within positions 2–6 of the MH, matches at these positions increase the probability of selection of an MH by Pol θ. To further understand how the length of terminal MH influences joining by Pol θ,

**Fig. 3 | Mismatched ends promote internal, imperfectly paired MH selection by Pol θ. a** Schematic of oligonucleotides Oligo 0N0M – 6N6M paired with Oligo 7. **b** Pol θ Δcen-mediated end-joining of paired oligonucleotides with a core 4 bp MH followed by 0–6 nt mismatches (panel **a**). Electrophoresis of 10 min end-joining reaction mixtures separated on a native 10% polyacrylamide gel (*n* = 2). The percentage of TMEJ products is labeled under the corresponding lanes. The intensity was measured with ImageJ. **c** Alignment of sequencing outcomes of Pol θ Δcen-mediated end-joining with Oligo 0N0M and Oligo 7. The blue box indicates the designed MH and the blue line distinguishes top and bottom strand TMEJ outcomes. A larger typeface version of the alignment is in Supplementary Fig. 8c. Major anchoring positions and possible pairing information of MHs are depicted. The

arrow indicates the direction of extension by Pol θ Δcen. Possible MHs are encompassed by a green square. Matches are labeled with light blue and mismatches with red. **d** Alignment of sequencing outcomes of Pol θ Δcen-mediated end-joining with Oligo 3N3M and Oligo 7. The blue box indicates the designed MH and the blue line distinguishes top and bottom strand TMEJ outcomes. A larger typeface version of the alignment is in Supplementary Fig. 8d. Major anchoring positions and possible pairing information of MHs are depicted. The arrow indicates the direction of extension by Pol θ Δcen. Possible MHs are encompassed by a green square. Matches are labeled with light blue and mismatches with red. Source data are provided as a Source Data file.

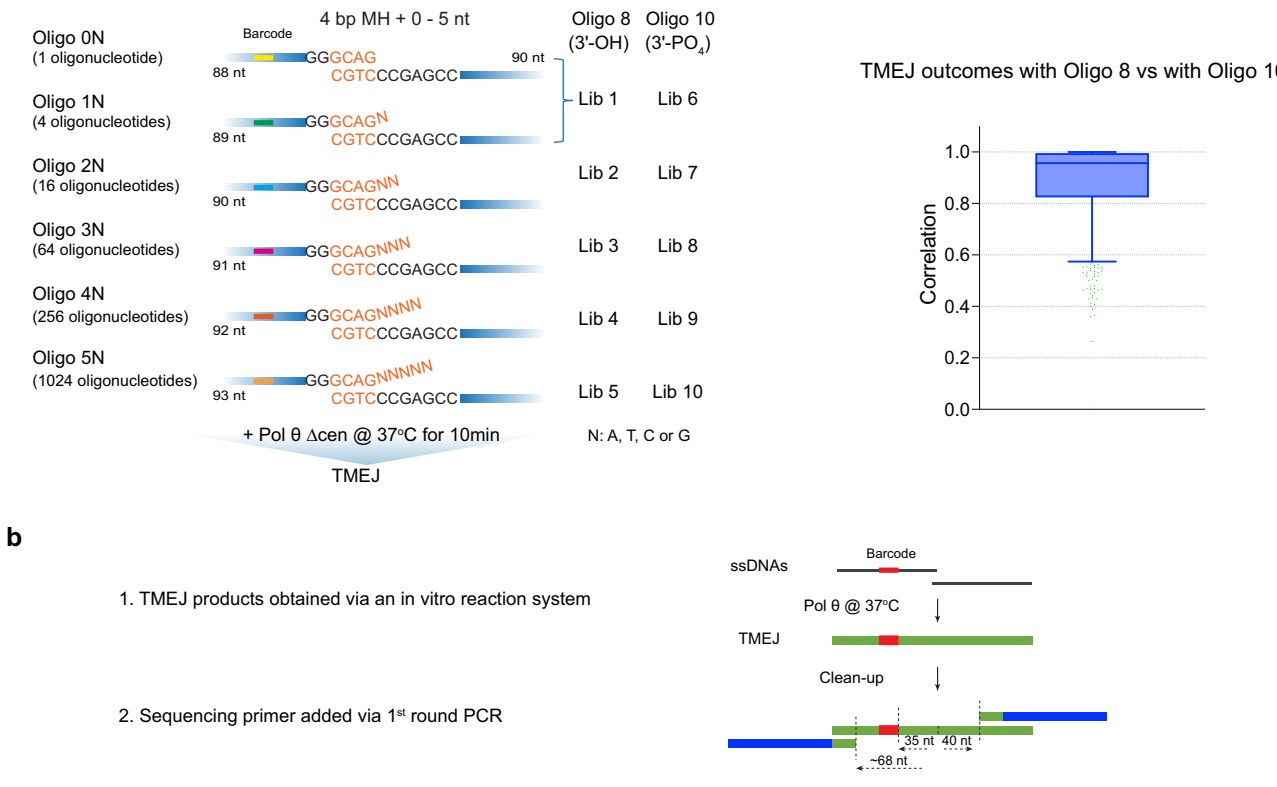

**Fig. 4 | Strategy for high-throughput end-joining sequencing analysis.**
**a** Designed oligo pools for high-throughput end-joining sequencing analysis. The core 4 bp MH are followed by 0–5 random nucleotides (N). A specific barcode (colored box) was designed for each oligonucleotide (Supplementary Data 1). Pools were paired with two different bottom oligonucleotides. Oligo 8 is the unblocked bottom oligonucleotide with 3′ OH. Oligo 10 is the blocked bottom oligonucleotide with 3′ PO₄. The TMEJ reactions were performed at 37 °C for 10 min. Lib 1–10 were then prepared using TMEJ reaction products originating from the corresponding oligo pools and bottom Oligo 8 (Lib 1–5) and Oligo 10 (Lib 6–10). **b** Schematic of high-throughput sequencing sample preparation. (1) Extension products (green) are generated by the reaction of Pol θ with pairs of DNA oligonucleotides. (2) Sequencing primers are added in a first-round PCR. This allows 40 nt of product to

be uniquely analyzed on the right side (extension from the top strand) and 35 nt of product to be uniquely analyzed on the left side (when extension from the bottom strand is possible). (3) Adaptors and indexes are added with the second round of PCR. (4) Amplicons are pooled for paired-end sequencing (NovaSeq6000). Further details are in the Methods. **c** Cosine correlation between end-joining deletion outcomes of oligo pools with unblocked bottom oligonucleotide (Oligo 8) and outcomes of oligo pools with blocked bottom oligonucleotide (Oligo 10). Totally 1365 individual oligonucleotide deletion outcomes were compared. The indistinguishable outcomes extended from terminal MHs are not included in the correlation analysis. Box plots show the median (center line), the 25th and 75th percentiles (bounds of box), and Tukey whiskers (1.5X interquartile range). Source data are provided as a Source Data file.

we designed paired oligonucleotides with lengths of terminal MHs from 3 to 8 bp. Longer terminal MHs led to higher efficiencies of TMEJ (Supplementary Fig. 6g). For comparison, results were extracted from the high-throughput data for top strands with terminal matched MHs

of 4–9 bp. This showed that longer potential terminal MHs are indeed used by Pol θ with high efficiency (Supplementary Fig. 6h). Once the terminal 5–6 bp are matched for this relatively GC-rich sequence, additional matching does not improve the efficiency.

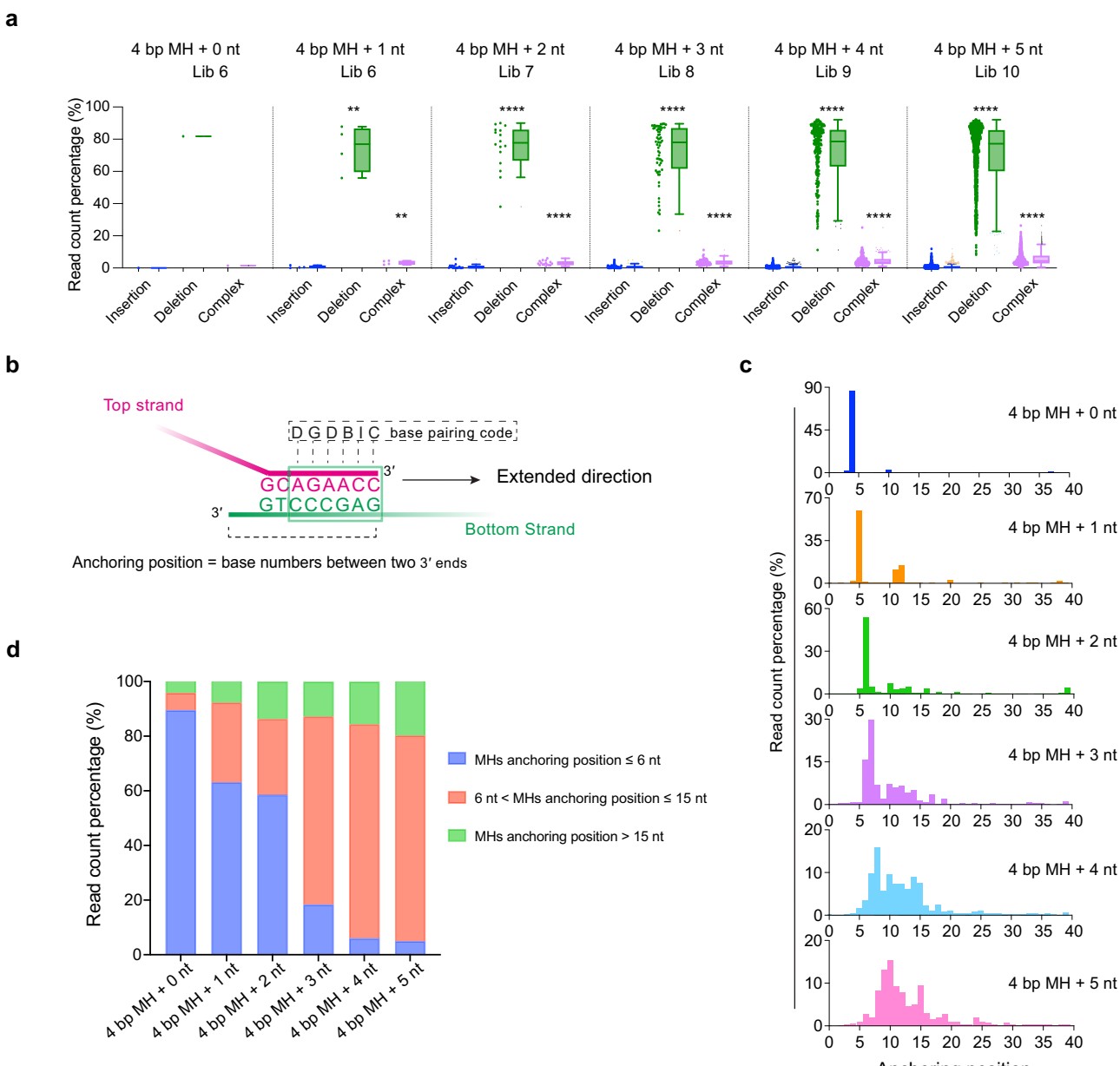

**Fig. 5 | MHs are usually selected by Pol θ within 15 nucleotides of the 3′ end of ssDNAs. a** End-joining read count percentage for insertions, deletions and complex event outcomes for the indicated top oligo libraries with a blocked bottom oligonucleotide (Oligo 10). Each top oligo library (4 bp MH + 0–5 nt) contains 1, 4, 16, 64, 256, and 1024 individual oligonucleotide outcomes, respectively. Dots indicate the individual read counts, and the box plot shows the distribution. Box plots show the median (center line), the 25th and 75th percentiles (bounds of box), and Tukey whiskers (1.5X interquartile range). The statistical analysis was performed using a paired two-tailed t-test. Significance is indicated as follows: ns (not significant, $p > 0.05$), *($p ≤ 0.05$), ** ($p ≤ 0.01$), *** ($p ≤ 0.001$), and **** ($p ≤ 0.0001$). **b** Definition of MH anchoring position for the extended top oligonucleotide and bottom oligonucleotide template. Arrow indicates direction of Pol θ extension. The anchoring position was defined as the distance (number of bases) between the 3′ ends of the joining ssDNAs. The assigned base pairing code for this example is labeled. Codes for all pairing types are described in Supplementary Data 1. A 6 nt MH region is boxed in a green square. **c** MH anchoring patterns of different oligo pools with a blocked bottom oligonucleotide (Oligo 10). Read count percentages were plotted with anchoring positions. **d** Read count percentage of MHs anchored at different regions for the indicated oligo libraries with a blocked bottom oligonucleotide (Oligo 10). Source data are provided as a Source Data file.

## Pol θ usually extends one single-stranded oligonucleotide when no terminal MH is present

The sequencing data provide the opportunity to determine whether both strands are used equally by Pol θ at the beginning of end-joining. As noted, blocking one end extension did not inhibit extension of the other strand by Pol θ (Fig. 2), or have a large effect on repair outcomes (Fig. 4c). It is possible, however, that extension of both strands might extend in coordination when there are no blocked termini. Using data with the unblocked bottom strand, we compared the frequencies of top oligonucleotide pool outcomes and bottom strand outcomes and

found that the two strands are not extended at equal frequencies by Pol θ (see percentage in Fig. 7a and read counts in Supplementary Fig. 7). Bottom strand outcomes are more frequent than top strand outcomes, consistent with results of small-scale experiments (Fig. 3d and Supplementary Fig. 3e). After filtering out erroneous reads and insertions, deletion outcomes were further analyzed for MH anchoring positions. Increasing diversity of the top strand pool resulted in more MH anchoring positions, while the preferred MH anchoring positions of the fixed bottom strand did not change (Fig. 7b). This analysis suggests that bottom strand MH anchoring and extension is

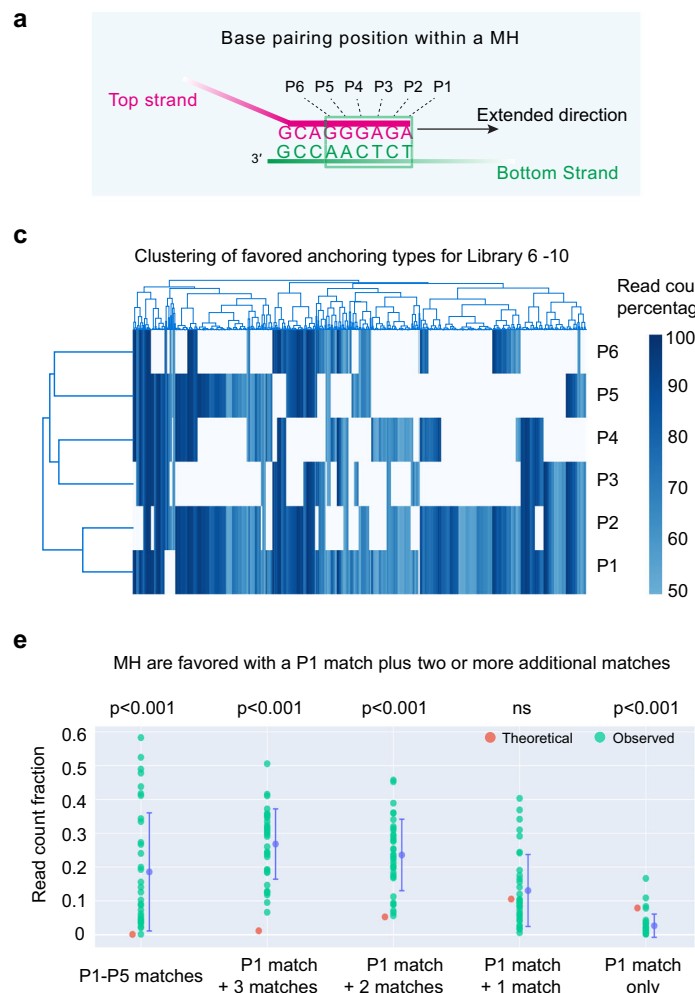

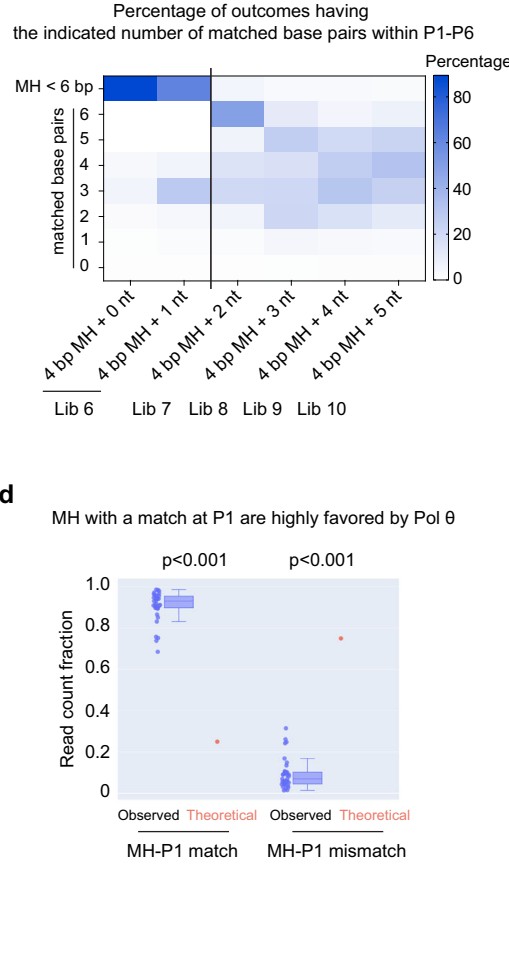

**Fig. 6 | MH selected by Pol θ usually contains mismatches. a** Diagram of base pairing positions P1–P6 of a MH. In this example, P1–P4 are matched, and P5–P6 are mismatched. **b** Heatmap showing read count percentage for each MH type utilized by Pol θ Δcen in the indicated libraries prepared with different oligo pools and a blocked bottom oligonucleotide (Oligo 10). MH <6 bp means the MH is anchored within 5 nucleotides of the 3' end of the bottom strand. MH matched base pairs (0–6) indicates that the MH contains 0–6 matched bases. **c** Cluster plot of the favored MH used by Pol θ Δcen for end-joining reactions. All anchoring positions for Library 6–10 oligonucleotide pools (blocked bottom "Oligo 10") were analyzed. If an oligonucleotide used the same MH in more than 50% of the read counts, it was designated a favored MH. Each column corresponds to one of these 910 individual oligonucleotides having a favored MH. Rows correspond to positions P1 to P6 of the MH as in panel (**a**). The percentage of matching at each position in the favored MH

is indicated with a *blue* color scale. Mismatched positions are white. **d** Matching at position P1 of an MH is favored by Pol θ for anchoring and extension during initiation of end-joining. The fraction of matches or mismatches at 35 P1 positions (blue dots and box plot) is compared with theoretical fractions (red dots). Box plots show the median (center line), the 25th and 75th percentiles (bounds of box), and Tukey whiskers (1.5X interquartile range). The statistical significance is labeled with the *p* value derived from a unpaired two-sided *t*-test. **e** MH containing >2 matches show higher anchoring frequencies than theoretical frequencies. Experimental anchoring frequencies of five different types of MHs at 35 positions are shown with green dots and the mean ± standard deviation in blue. The comparison with theoretical frequency (red dots) was done with a two-sided *t*-test. The resulting *p* value is shown; ns indicates no significant difference. Source data are provided as a Source Data file.

independent of top strand MH anchoring and extension during end-joining in vitro.

## Interrupted MH occurs in vivo

Pol θ can use information within at least 6 bp to select an MH, and this concept may help in understanding the MH identification during TMEJ in cells. To show the potential of such analysis, we re-analyzed data from a comprehensive report[12] that examined double-strand break repair in human K562 cells. This study determined repair outcomes after the introduction of a DSB at four different CRISPR-Cas9 target sites in a sequence from the human *HBB* gene. We analyzed all 19 deletions that were scored as highly dependent on POLQ activity, as determined by CRISPRi-mediated *POLQ*-knockdown (Table 1). By the standard definition, these deletions were interpreted as using short (0–4 nt) microhomologies (Table 1). Reanalysis shows that 14 of the 19

deletions could have arisen by use of longer, interrupted microhomologies.

For example, at DSB target site 3, there were frequently occurring 10 bp deletions dependent on Pol θ (Table 1). Using the standard definition for MH, these were scored as two distinct events with different deletion borders, one initiated at a 1 bp MH and one at a 2 bp MH. A simpler interpretation (Fig. 8a) is that both arose from the same MH pairing, a 6 bp interrupted MH (CXXXAA), and that the deleted segment is identical for both events. Following the completion of TMEJ, unpaired bases will be resolved by mismatch repair or a cycle of DNA replication. This yields the two events that were originally scored, by the standard definition of MH, as independent deletions.

At DSB target site 2, two major Pol θ-mediated repair outcomes were 13 bp deletions, initially scored as arising from distinct 1 bp and 4 bp MH sites (Table 1). An alternative interpretation is that both arose

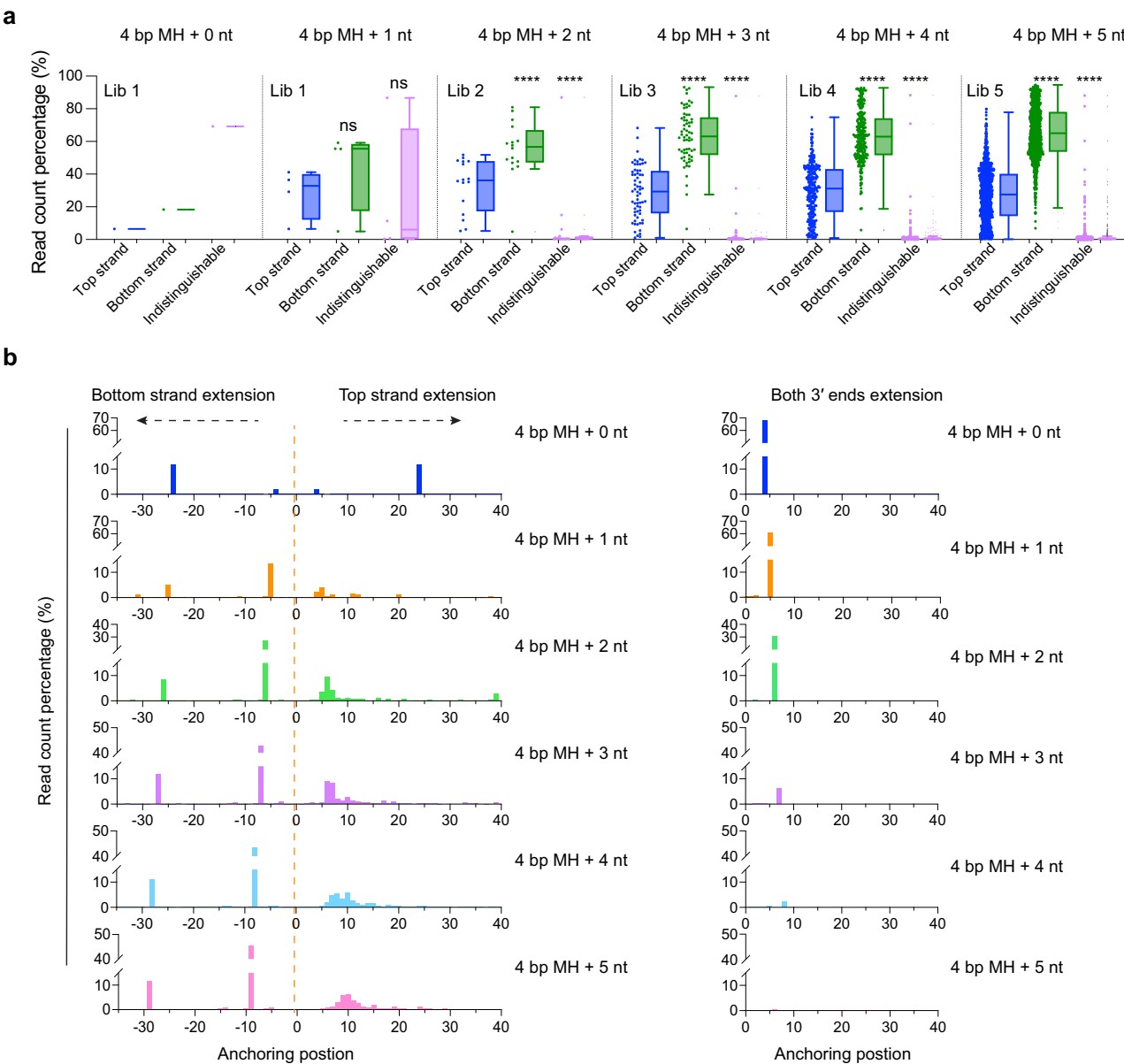

**Fig. 7 | Pol θ usually extends one single-stranded oligonucleotide during end-joining. a** End-joining by Pol θ Δcen, showing read count percentage of top strand outcomes, bottom strand outcomes and indistinguishable outcomes for the indicated libraries prepared with different oligo pools and unblocked bottom oligonucleotide (Oligo 8). Each library outcome (4 bp MH + 0 – 5 nt) contains 1, 4, 16, 64, 256, and 1024 individual oligonucleotide outcomes, respectively. Outcomes are classified as "indistinguishable" if they can arise by extension of either top or bottom strands. Dots indicate the individual read counts, and the box plot shows the distribution. Box plots show the median (center line), the 25th and 75th percentiles

(bounds of box), and Tukey whiskers (1.5X interquartile range). The statistical analysis was performed using a paired two-tailed *t*-test. Significance is indicated as follows: ns (not significant, $p > 0.05$), * ($p \leq 0.05$), ** ($p \leq 0.01$), *** ($p \leq 0.001$) and **** ($p \leq 0.0001$). The indistinguishable group was most enriched in Lib 1, as this library includes terminal 4 bp and 5 bp MH (see also Supplementary Fig. 7). **b** MH anchoring positions for deletions arising from different libraries prepared with the designated oligo pools and unblocked bottom oligonucleotide (Oligo 8). Read count percentages are plotted for each MH anchoring position. Source data are provided as a Source Data file.

from the same MH pairing, a 7 bp interrupted MH (TGGCXXA), yielding the same deleted segment for both events (Fig. 8b). Replication or mismatch repair would give rise to the two outcomes.

At DSB target site 1, two distinct 4 bp deletions were categorized as arising from distinct 2 bp and or 0 bp MH sites (Table 1). An alternative interpretation is that both arose from the same MH pairing, a 4 bp interrupted MH (XAGX), with the same deleted segment for both events (Fig. 8c). Additionally, a 21 bp deletion was previously classified as arising from 4 bp MH by the standard definition (Table 1), but is more simply explained as arising from a longer, more stable interrupted MH AGTGXXGGCTG, Fig. 8d). Initiating repair at this site also

accounts for a second 21 bp deletion identified in the dataset[12] (Fig. 8d).

We also analyzed results from a study of repair of a CRISPR-Cas9 DSB in the human *HPRT* gene[20]. We focused on deletions that were inferred to use short microhomologies and occurred only in experiments with *POLQ*-proficient U2OS cells, but not in *POLQ* knockout cells. One such event is an 11 bp deletion with a reported 2 bp TC microhomology (Supplementary Fig. 10a). Removal of the unpaired bases in the top strand, extension, gap filling and replication or mismatch repair would result in the 11 bp deletion. This event was scored as a 2 bp MH by the traditional definition, but it would be scored as a

**Table 1 | Analysis of microhomologies (MH) used for deletions at DSB target sites[a]**

| Figure | Deletion size | Deletion sequence and cleavage site[b] | Previously[a] identified MH | Redefined MH[c] |
|---|---|---|---|---|
| | | DSB target site 1: CCAGTGCAGGCTGCCTATCAGA ▽ AAGTGGTGGCTGGTGTGG | | |
| Fig. 8c | 4 bp | AGA ▽ A | AG | XAGX |
| Fig. 8d | 21 bp | AGTGCAGGCTGCCTATCAGA ▽ A | AGTG | AGTGXXGGCTG |
| | 16 bp | AGGCTGCCTATCAGA ▽ A | AG | AG |
| | 7 bp | GA ▽ AAGTG | G | AXXXG |
| Fig. 8c | 4 bp | CAGA ▽ | 0 | XAGX |
| | | DSB target site 2: GGTCTGTGTGCTGGCCTATC ▽ ACTTTGGCAAAGAATTCACCCC | | |
| Fig. 8b | 13 bp | TGGCCTATC ▽ ACTT | TGGC | TGGCXXA |
| Fig. 8b | 13 bp | ATC ▽ ACTTTGGCAA | A | TGGCXXA |
| | 6 bp | CTATC ▽ A | CT | CTXTG |
| | 18 bp | CTGTGTGCTGGCCTATC ▽ A | CT | CTXTG |
| | | DSB target site 3: CCTGGCCCACAA ▽ GTATCACTAAGCTCGCTT | | |
| Fig. 8a | 10 bp | CCACAA ▽ GTAT | C | CXXXAA |
| Fig. 8a | 10 bp | AA ▽ GTATCACT | AA (or AAG)[d] | CXXXAA |
| | 16 bp | CCTGGCCCACAA ▽ GTAT | C | TXXC |
| | 9 bp | CACAA ▽ GTAT | CAC | CACXA |
| | 9 bp | A ▽ GTATCACT | A | CACXA |
| | 7 bp | CAA ▽ GTAT | CA | CA |
| | 4 bp | AA ▽ GT | A | A |
| | | DSB target site 4: CCTGGCCCACA ▽ AGTATCACTAAGCTCGCTT | | |
| | 7 bp | CA ▽ AGTAT | CA | CA |
| | 9 bp | CACA ▽ AGTAT | CAC | CAC |
| | 16 bp | CCTGGCCCACA ▽ AGTAT | C | AXTXXC |

[a]Target sites, deletion size and microhomologies from Fig. 6D and Supplementary Fig. 4 of ref. 12.
[b]The Cas9 cleavage site is marked by ▽.
[c]X indicates a mismatch in the predicted interrupted microhomology, see Fig. 8 for examples.
[d]Identified as microhomology AAG in ref. 12, but likely intended as AA because the G lies on the opposite side of the Cas9 cleavage site.

6 bp MH with two internal mismatches by the revised definition emerging from this study.

A straightforward case can be found in the same dataset where a 9 bp deletion is produced from a 3 bp terminal MH, consistent with an event where the top strand is extended from the 3 bp MH (Supplementary Fig. 10b).

Another example of mismatched MH is provided by a 181 bp deletion scored as occurring with 0 bp microhomology. Reconsideration of this MH shows that two base pairs at positions P3 and P4 flanked by mismatches on either side could have contributed to MH selection (Supplementary Fig. 10c). After processing of the bottom strand by nuclease activity and subsequent extension, the product could be replicated or subject to DNA mismatch repair. Outcomes would be scored as a 0 bp MH by the traditional definition, but as a 2 bp MH by the revised definition emerging from the present study

## Discussion

Experiments in various organisms suggest that TMEJ is usually initiated at short (2–6 bp) MHs[2,4,9]. A Pol θ-dependent process identifies these MHs in the two single-stranded 3′ tails formed after resection at a DSB. However, it has been puzzling that some Pol θ-dependent repair events appear to have no obvious MH, or only 1 bp. It has also not been clear whether Pol θ itself is sufficient to initiate priming at internal short MH, or whether additional protein factors are needed. A further question has been whether subsequent extension by Pol θ must use both ends of the break. Finally, is a microhomology defined by a sequence of contiguous bases, or are mismatches permitted within an MH that is selected by Pol θ? Answers to these questions have emerged from the present study.

Limitations in our understanding of the above questions are due in part to the low diversity of end-joining substrates that have been tested. Traditional MH analysis has typically been based on end-joining of specific extra-chromosomal substrates or DNA double-strand breaks introduced at specific sites in a chromosome. Most in vitro end-joining analyses use terminally matched MHs, which are unlikely to occur at random DSB sites. Here, high-throughput sequencing with large pools of oligonucleotides containing diverse 3′ ends enabled us to unambiguously profile MH outcomes. Our analysis shows that purified Pol θ has an intrinsic ability, without additional protein factors, to locate and extend from internal MHs. Such internal MH were preferred, even if a 4 bp MH was present within a few bp of the 3′ terminus. Around 80% of these MH are within 15 nt of the 3′ end of the ssDNA (Fig. 5d, Libraries 9 and 10). This is consistent with cellular studies showing a similar distribution of internal MH positions[5]. Because Pol θ locates a MH close to a ssDNA end, this restricts the size of deletions formed during TMEJ.

In cells, MH joining sites are often selected at positions where both strands originally had a few nucleotides extending beyond the MH site[5,6]. To allow extension in these cases, at least one strand is shortened by the 3′ to 5′ exonuclease activity of Pol δ[13]. The results reported here are also applicable to this general case. If no appropriate MH can be extended by Pol θ, a single-stranded terminus can be shortened by Pol δ exonuclease. Pol θ will then use the shortened single-strand in a search for a new set of anchoring positions. In vivo, there are additional factors that might modulate TMEJ outcomes. For example, ssDNA is initially bound by RPA, which can be displaced by the ATPase activity of Pol θ[15,21]. ATP is another potentially modulating factor, but we show here that the intrinsic ATPase activity of Pol θ did not affect the MH selection pattern for end-joining.

We found that blocking the extension of one oligonucleotide end with a 3′ phosphate did not affect anchoring of the other strand at an MH and extension by Pol θ. This indicates that

**a**  MH = CXXXAA

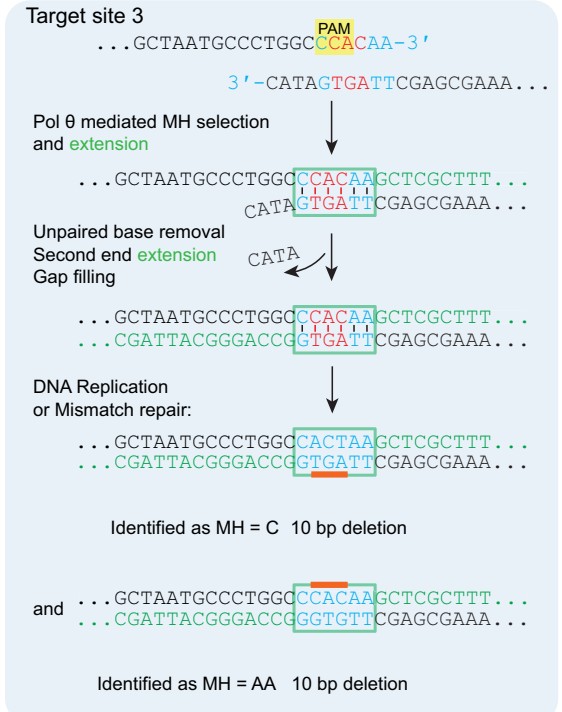

**b**  MH = TGGCXXA

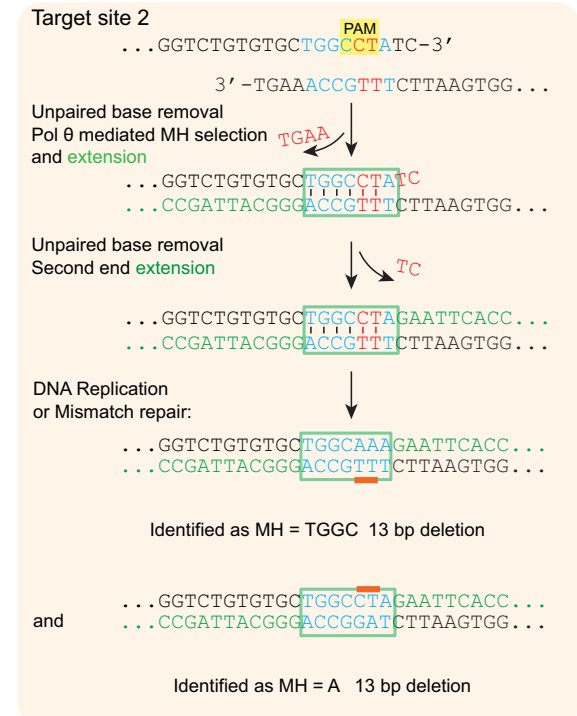

**c**  MH = XAGX

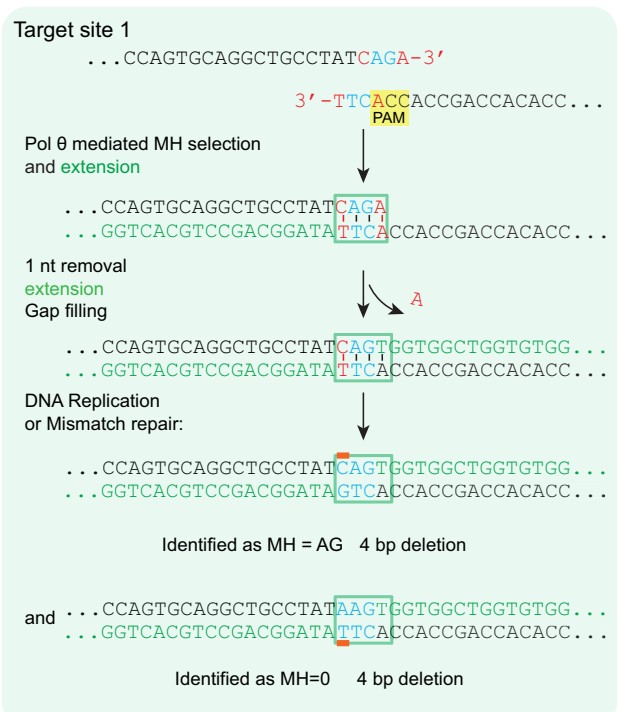

**d**  MH = AGTGXXGGCTG

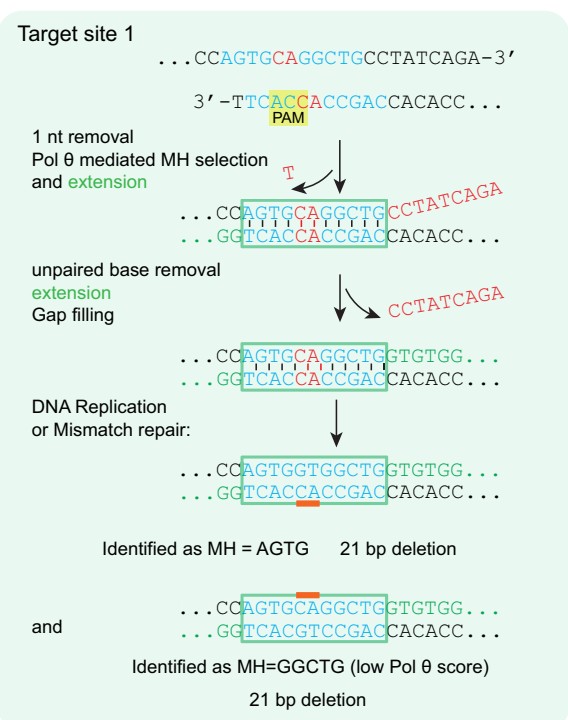

extension of both strands is not necessary to initiate TMEJ. Consistent with this, the fractions of outcomes assigned to the two strands are significantly different. A study where pre-resected substrates were transfected into cells showed that Pol θ-mediated end-joining is blocked when both ends have at least three mispaired nucleotides containing nuclease-blocking phosphothioate bonds[13]. Fully normal levels of repair were observed when only

one of the two ends was blocked in this manner. This supports the concept that only one oligonucleotide strand needs to be extended by Pol θ to initiate TMEJ. The remaining unprocessed strand retains a non-homologous tail. This could then be processed by single-strand break repair nucleases to complete repair (Fig. 9). Thus, TMEJ can result from the successful priming of only one strand.

**Fig. 8 | Pol θ-dependent repair is initiated at interrupted MH in vivo.** Examples of proposed mismatched MH used to generate *POLQ*-dependent deletions, derived from analysis of DSB repair at four CRISPR-Cas9 targets in a sequence derived from the human *HBB* gene[12]. Target sites, deletion size and microhomologies derived from ref. 12 are summarized in Table 1 for these examples and the full *POLQ*-dependent dataset. At the top of each panel, the local sequence environment is shown for the two 3′ single-stranded DNA ends to be joined. Matched bases in the MH are colored blue, and mismatched bases are colored red. Flanking unpaired bases are removed by nuclease action[13]. Extension by Pol θ is indicated by bases colored green. The internally mismatched bases will be resolved in the cell by mismatch repair or DNA replication, yielding the two outcomes shown at the bottom of each panel. **a** In target site 3, the same internally mismatched MH, with 3 paired bases, can give rise to two outcomes. These were previously scored as two different 10 bp deletions with MH lengths of 1 or 2. **b** In target site 2, the same internally mismatched MH, with five paired bases, can give rise to two outcomes. These were previously scored as two different 13 bp deletions with MH lengths of 1 and 4. **c** In target site 1, the same internally mismatched MH, with two paired bases, can give rise to two outcomes. These were previously scored as two different 4 bp deletions with MH lengths of 0 and 2. **d** In target site 1, the same internally mismatched MH, with nine paired bases, can give rise to two outcomes. These were previously scored as a *POLQ*-dependent 21 bp deletion with MH length of 4, and a *POLQ*-independent 21 bp deletion with MH length of 5.

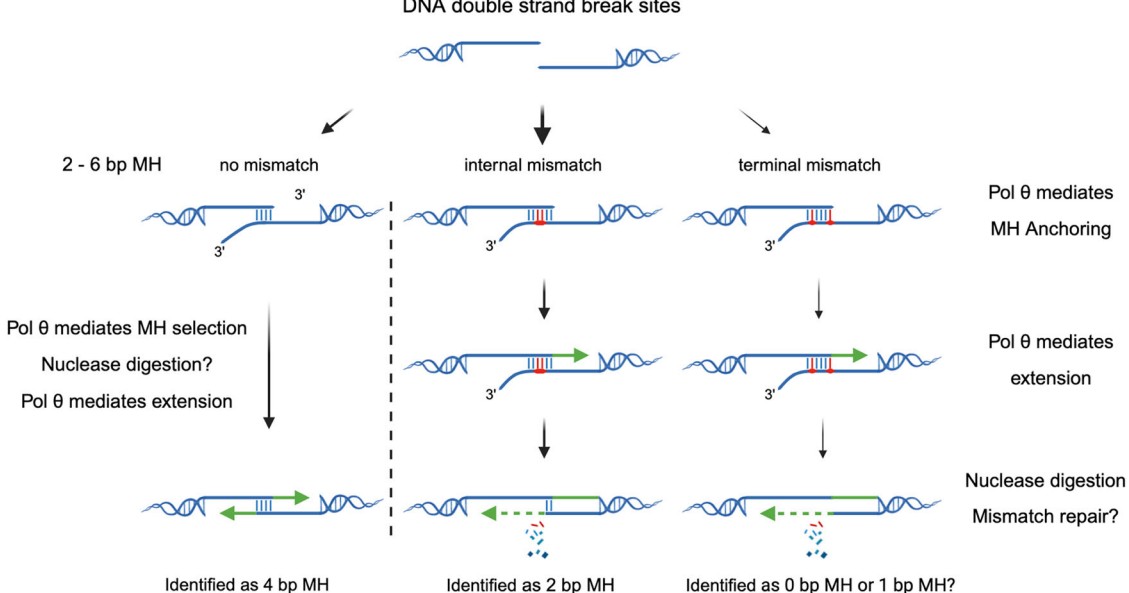

**Fig. 9 | Consequences of selection by Pol θ of an MH containing mismatches during TMEJ.** The left pathway shows that when Pol θ selects an MH without mismatches (shown as 4 bp as an example), overhanging ends will be removed if needed, and the final MH will be identified as 4 bp. The middle pathway shows that an internal MH may be selected that includes mismatches (at P3 and P4 in the example shown). After processing of 3′ overhanging ends, and DNA replication or subsequent mismatch repair, the MH will be identified as 2 bp, even though it used pairing information from a 6 bp window. The right pathway shows that similarly, a terminal MH may be selected that includes mismatches (at P1 and P5 in the example shown). After processing of 3′ overhanging ends, DNA replication or mismatch repair can occur. The MH may then be identified as 0 or 1 bp even though it used pairing information from a 6 bp window. Created in BioRender. Li, Y. (2025) https://BioRender.com/vxre3ij.

Our analyses reveal some key determinants of MH selection by Pol θ. We focused on the potential for base pairing in the last 6 nt of a primer because efficiency does not improve after MH are longer than ~6 bp, and the polymerase domain of Pol θ has specific contacts with the phosphodiester backbone for the last 6 nt of the primer[19,22].

Three important conclusions arose from the high-throughput DNA sequencing analysis. First, pairing at terminal position P1 is the most important determinant for MH selection by Pol θ. Second, selected microhomologies favor additional matching at P2–P6. Third, the matched bases need not be consecutive, and thus mismatches are tolerated within P1–P6. This is consistent with the known ability of purified Pol θ to tolerate some base pair mismatches between primer and template, though often with lower efficiency of priming[8,11]. After Pol θ extends from MHs with internal mismatches, the resulting primer-template mismatch would be a substrate for a subsequent round of DNA replication, or for DNA mismatch repair. The strand chosen for correction by the mismatch repair system might be randomly chosen or marked as a newly replicated strand. Depending on the final repair outcome, it might not be obvious that a mismatched priming site was initially chosen (Fig. 9).

Our results suggest that for TMEJ, a broader definition of MH is needed than is normally used. This is because non-consecutive base pairing at P1–P6 can contribute to MH selection. For example, a MH may be identified as only 1 or 2 bp by the typically accepted definition of MH requiring consecutive bases, even though more bp contributed to the MH selection (Fig. 8 and Supplementary Fig. 10).

Homologous recombination-defective cancers are characterized by insertion-deletion mutation signatures associated with short microhomologies[23]. Such signatures have some correlation with the action of Pol θ[20]. Further analysis of mutations in homologous recombination-defective cancers, taking account of potential mismatched microhomologies, may help better define such insertion-deletion signatures to identify cancer mutations that involve TMEJ.

## Methods

### Proteins and oligonucleotides

Human DNA Polymerase θ Δcen sequence was inserted into phCMV1-2XMBP[24] with optimized mammalian expressed codon and a short linker GSAGSAAGSGEF in place of central domain residues 911–1791 (GenScript). DNA was transfected into human Expi293F cells. After 48 h, cells were harvested and protein was purified as described[25]. Klenow Fragment (3′ → 5′ exo-) (M0212L) was purchased from New England Biolabs.

All designed oligonucleotides and oligonucleotide pools (sequences listed in Supplementary Table 1 and Supplementary Data 1, 2) were synthesized with or without fluorescent labeling by Integrated DNA Technologies.

## Analysis of TMEJ by electrophoresis

About 50 nM Cy5 and 50 nM Cy3 5′-labeled single-stranded oligonucleotides were incubated with 100 nM Pol θ Δcen at 37 °C in 20 μL reaction mixtures (25 mM potassium phosphate pH 7.0, 0.1 mg/mL bovine serum albumin (BSA), 5 mM dithiothreitol (DTT), 5 mM MgCl$_2$ and 100 μM dNTPs) for different time points as indicated. The reaction was stopped with 4 μL 6X stop buffer (300 mM Tris-HCl pH 7.5, 3 mg/ml proteinase K (P8107S, New England Biolabs), 120 mM ethylenediaminetetraacetic acid (EDTA) and 1.2% sodium dodecyl sulfate (SDS)) for another 15–30 min at 37 °C. For the reactions with ATP, oligonucleotides mixed with 5 mM ATP were incubated with Pol θ Δcen at 37 °C in 20 μL reaction mixtures for 15 min. Reactions were mixed with 5 μL 6X native gel loading buffer (40% sucrose with 0.025% bromophenol blue), loaded onto native 10% polyacrylamide gel (10% 19:1 acrylamide:bis-acrylamide, 0.5 X TBE (Tris-borate-EDTA) buffer, 0.075% ammonium persulfate (APS) and 0.0375% tetramethylethylenediamine (TEMED)) and run at 5 Watt/gel using Bio-Rad PROTEAN II XL Gel Running System. Fluorescence was detected by scanning gels with an Amersham Typhoon 5 (Cytiva).

TMEJ bands were confirmed via digestion with BamHI-HF (R3136S, New England Biolabs) or SacI-HF (R3156S, New England Biolabs). Before adding the stop buffer, 20 μL mixtures were added with 2.2 μL 10X Cutsmart buffer (B6004S, New England Biolabs) and 1 μL restriction endonucleases and incubated at 37 °C for an extra 1 h. Remaining steps are the same as above.

## TMEJ product measurement

ImageJ (https://imagej.net/ij/index.html) was used to calculate the percentage of TMEJ products on native polyacrylamide gels. The white bands and adjacent smear were considered TMEJ products. The percent TMEJ was calculated independently for the Cy5 and Cy3 channels, comparing the intensity of TMEJ products to the total intensity of all the bands in each lane. The average of the TMEJ percentages for the two fluorescence signals was calculated as the total TMEJ percentage. The plot was made using Prism 10 (GraphPad, https:// www.graphpad.com).

## Small-scale sequencing of TMEJ outcomes

About 40 μL TMEJ reaction mixtures were prepared with different paired oligonucleotides and stopped with 8 μL 6X stop buffer for another 15–30 min at 37 °C. For reactions with oligonucleotide containing dUTPs, 1 μL of USER enzyme (M5505S, New England Biolabs) was added to the reaction mixture and allowed to incubate for an extra 30 min to remove dUTPs, before the reactions were stopped. Reaction products were subsequently cleaned up with the MinElute PCR purification Kit (28004, QIAGEN). TMEJ products were amplified using NEBNext® UltraTM II Q5® Master Mix (M0544L, New England Biolabs) and primers R1/F1 for 10-19 cycles. After gel cleanup with QIAquick Gel Extraction Kit (28004, QIAGEN), the concentration of the amplicons was measured. Equal amounts of each amplicon were then used for assembly into a pUC19 vector via NEBuilder® HiFi DNA Assembly Master Mix (E2621L, New England Biolabs). The assembled vectors were transformed into homemade competent E. coli TOP10 cells and spread on a selection plate. After incubation overnight at 37 °C, 50 colonies were sent to Genewiz (South Plainfield, NJ) or MCLAB (South San Francisco, CA) for Sanger sequencing with primer F-Seq. Sequencing results were analyzed by alignment with the reference sequence via MacVector v18.5 (https://www.macvector.com).

## Library preparation for high-throughput sequencing of TMEJ outcomes

For high-throughput sequencing experiments, six different oligonucleotide pools were designed with terminal 0–5 random nucleotides following a 4 bp microhomology which can pair with the 3′ end of a "bottom strand" oligonucleotide (Fig. 4a). To track the products of each reaction, a unique barcode was included in each oligonucleotide (Supplementary Data 1). In detail, 40 μL TMEJ reaction mixtures contained different oligonucleotide pools and a specific bottom strand. Reaction mixtures were incubated with Pol θ Δcen at 37 °C for 10 min. The reactions were stopped with 8 μL 6X stop buffer, and the reaction products were subsequently cleaned up using a MinElute PCR purification Kit. About 20 μL eluate was obtained after cleanup and used as a template for the first round PCR (nine cycles) to add the sequencing primer. The reactions were assembled in 3 × 50 μL mixtures (1X NEBNext® UltraTM II Q5® Master Mix (M0544L, New England Biolabs) with 0.5 μM primers (ASP-ForL/ASP-RewL) and 2 μL eluate. To attach the Illumina adapter and indexes for each library (Supplementary Data 1), a second round of PCR (100 μL, six cycles) was conducted using 120 ng of the first round PCR purified products plus the Dual Index Primers Set 1 from NEBNext® Multiplex Oligos for Illumina (E7600S, New England Biolabs). QIAquick Gel Extraction Kit (28004, QIAGEN) was then used to obtain a 30 μL eluate of the reaction products containing the Illumina adapters. The concentration was measured and adjusted, and a 10 nM 100 μL premade library was pooled and quantified with the TapeStation system and sequenced on NovaSeq6000 by 100 nt paired-end sequencing with 20% PhiX addition at MDACC - Advanced Technology Genomics Core (ATGC).

## Bioinformatic analysis

After sequencing, the data were stored and analyzed using the HPC clusters at MD Anderson Cancer Center. More details are in Supplementary Methods. Briefly, each library and sample was demultiplexed based on specific indexes and oligonucleotide barcodes (Supplementary Data 1). Quality checking, merging, filtering, classification and analysis were conducted for all reads. Filtering was done to remove reads which are shorter than the original top strand sequence (<88 bp). Pairwise global alignment was used for further classification. When the library was prepared with a blocked bottom oligonucleotide (Oligo 10), after alignment with the reference sequence (joining sequence with two oligonucleotide substrates), query reads were classified into errors, deletions, insertions and complex events for further analysis. When the library was prepared with unblocked bottom oligonucleotide (Oligo 8), query reads could be classified into error unclassified, top strand outcomes, bottom strand outcomes and indistinguishable outcomes for further analysis. Deletions, the major outcomes of end-joining and sequencing, were further analyzed based on deletion size to determine the MHs anchoring position and pairing information. Statistical significance (two-sided $t$-test) were obtained using GraphPad Prism 10, which reports $p$ value ranges when $p < 0.0001$.

## Reporting summary

Further information on research design is available in the Nature Portfolio Reporting Summary linked to this article.

# Data availability

The raw fastq files of high-throughput sequencing generated in this study have been deposited in the NCBI's Sequence Read Archive (SRA) database with the accession number PRJNA1178638 [https://www.ncbi.nlm.nih.gov/bioproject/1178638]. The designed oligo pools with specific barcodes and demultiplex indexes are available in Supplementary

Data 1. Requests for materials should be directed to the corresponding authors. Source data are provided with this paper.

## Code availability
Demo Code has been deposited on the Code Ocean platform at https://doi.org/10.24433/CO.2045180.v1.

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

## Acknowledgements
For useful comments on the manuscript, we thank Dale Ramsden, Gaorav Gupta, Sylvie Doublié, and Mélanie Prodhomme. We thank April M. Averill and Sylvie Doublié of the Protein Expression and Purification Core, P01 CA247773, for purified Pol δ. We appreciate ongoing discussions with our colleagues, including Liang Zhang, Rongjie Fu, Kei-ichi Takata, and Eli Rothenberg. We are grateful for the support of the High Performance Computing for Research facility at the University of Texas MD Anderson Cancer Center. Our work was funded by National Institutes of Health grants P01 CA247773 (to R.D.W.) and R35 GM137927 (to H.X.), and by the J. Ralph Meadows Chair in Carcinogenesis Research (to R.D.W.). H.X. is a CPRIT Scholar in Cancer Research. The content is solely the responsibility of the authors and does not necessarily represent the official views of the National Institutes of Health.

## Author contributions
Y.L. performed experiments, conceptualized the study, analyzed data, and co-wrote the manuscript. N.K.D. and B.L. developed analysis pipelines, analyzed data, and contributed to writing. M.R. and A.T.G. performed experiments and contributed to the writing. W.H., D.C.-M. and H.X. contributed to experimental design and to the writing. R.D.W. conceptualized the study, analyzed data, and co-wrote the manuscript.

## Competing interests
The authors declare no competing interests.
