## [Transparent Peer Review file · Nature Communications]

Pol θ -mediated end-joining uses microhomologies containing mismatches

Corresponding Author: Professor Richard Wood

Version 0:

Reviewer comments:

Reviewer #1

(Remarks to the Author)

In the manuscript, Li and colleagues investigate the mechanisms of microhomology selection by Pol θ during DNA repair. The authors employ a combination of *in vitro* biochemical assays and high-throughput sequencing analyses to provide new insights into how Pol θ selects and utilizes microhomologies during the end-joining process.

This work is technically sophisticated, offering a single-base resolution analysis of the microhomology selection mechanism by Pol θ . The data quality is robust and meets the standard for publication in Nature Communications.

However, my main concern lies in the advancement of novelty compared to previous studies. This issue could be addressed by incorporating additional experiments to further strengthen the manuscript before publication.

Major concerns:

This study provides a more comprehensive analysis of Poltheta functions using NGS sequencing compared to previous biochemical studies (e.g., Ref. 9). However, the core findings on mismatch tolerance and minimal base-pairing requirements are similar to those previously reported. To enhance the novelty of the manuscript, the authors should include additional discussions on how the use of NGS data advances the understanding of Poltheta functions beyond conventional biochemical approaches. While this study demonstrates that mismatched ends promote internal, imperfectly paired microhomology selection by Poltheta *in vitro*, the molecular mechanism underlying this process remains unclear. Additional experimental approaches, such as structural analysis, systematic biochemical assays, and cellular experiments, would be beneficial to elucidate how mismatched ends mechanistically influence internal anchoring by Poltheta.

Additionally, statistical analysis should be conducted for all quantitative data to ensure the robustness and reliability of the findings.

Reviewer #2

(Remarks to the Author)

In this work, Li and colleagues leverage an *in vitro* system to characterize the mechanisms by which DNA polymerase theta selects and aligns microhomologies (MHs) during TMEJ. Using a purified form of pol theta with a shortened linker domain, they systematically test libraries of oligonucleotides in end-joining reactions, assessing the importance of microhomology length, imperfect pairing, and non-homologous tails. From their analysis, they draw three important conclusions: (1) pol theta has a strong preference for MHs that are located with 15 nucleotides of the ssDNA ends; (2) pol theta often aligns MHs that contain mismatches, and ssDNA with multiple, non-contiguous mismatches can be substrates for joining; and (3) pol theta can utilize its polymerase activity to extend one annealed strand independently of the other.

Many cellular and organismal studies have shown that pol theta prefers to join MHs that are located close to the 3' ends of breaks; conclusion 1 validates these. Others have also suggested that MMEJ/TMEJ might involve alignment of MHs with mismatches, but to my knowledge this is the first rigorous experimental demonstration of this. The same is true for the third conclusion listed above, and the authors may want to stress this novelty. Taken together, the findings from this study highlight the unusual flexibility of TMEJ and further expand the range of end-joining repair events that might be attributed to

this error-prone repair process. If the findings are generalizable to more physiologically relevant contexts (see below), many studies that have assigned double-strand break repair products to either NHEJ, MMEJ, or TMEJ might need to be reinterpreted.

However, there are some weaknesses that need to be addressed to increase confidence that these data could represent the actual TMEJ mechanism in a cellular context. First, although the authors are using purified pol theta containing both helicase and polymerase domains, they don't include ATP in the end-joining reactions. It is possible that the helicase/ATPase domain of pol theta could unwind stem-loop extension products or disrupt poorly annealed MHs in an ATP-dependent manner. In addition, as the authors mention in the discussion, ssDNA is usually bound by RPA *in vivo* and needs to be stripped via pol theta helicase activity to promote efficient MH annealing. To investigate these possibilities, experiments that are critical to the conclusions in figures 1-3 should be repeated with ATP (at a minimum) and purified RPA if possible.

Second, in many of the reactions the use of internal, imperfectly paired MHs to promote end joining is necessitated due to the presence of non-homologous 3' tails. The Ramsden lab has shown that these tails can be removed through exonuclease activity of pol delta. If this occurs, and MH annealing is transient, it seems possible that some of the mismatched MHs might be disrupted by either pol delta exonuclease action or pol theta helicase activity, leading to a different spectrum of end-joining products than what was observed in these studies. Ideally, this should be tested using purified pol delta, but perhaps the exonuclease activity of pol delta is so robust that the ssDNA gets rapidly and non-specifically cleaved *in vitro* (Fig. S4)? Regardless, this caveat needs to be included in the discussion.

Other minor points:

1. The distribution of MH anchoring positions in Fig. 3f doesn't match the conclusion drawn from the data in Fig. 5. Is this because the MH anchoring position is being defined differently in Fig. 3? Similarly, why do the anchoring positions in Fig. 5 only extend to 40 nt to the right/left of the 3' ends? From Fig. 4b, it appears that there are 68 nt to the left of the ssDNA ends that are being sequenced. This needs to be clarified.
2. Fig. 6c: We had difficulty understanding how this clustering diagram demonstrates that matched base pairs are not usually consecutive. If they were consecutive, would each column be expected to contain an unbroken blue bar that spans at least part of the P1-P6 space? A more complete explanation here would help.
3. For consistency, the percentage of TMEJ products should be shown below the gels in Fig. 2A&C.
4. Figure 2 legend title: The use of the word 'terminal' is somewhat misleading here, since oligo 2N2M has a 2-base pair mismatched 3' terminus. We suggest using 'core MH,' as was used at the bottom of page 7.

Reviewer #3

(Remarks to the Author)

Reviewer #4

(Remarks to the Author)

DNA polymerase theta contributes to double-strand break (DSB) repair by extending single stranded DNA ends imperfectly paired at regions of limited homology. This study characterized Pol theta sequence preferences using an *in vitro* biochemical system coupled with high-throughput sequencing. The authors demonstrated that Pol theta can extend from terminal mismatches and then characterized its patterns of sequence usage. They observed that a match at the 3'-most nucleotide contributes the most to microhomology site selection, that end-joining occurs more frequently at regions with larger numbers of matched basepairs (up to six), and that mismatches within the six nucleotide microhomology region are allowed. Using their findings, the authors reinterpreted published data from an *in vivo* DSB repair assay done in human cells.

The article addresses the important mechanistic question of how Pol theta selects regions of microhomology, which are only a few bp in length. However, the context of the study is not fully explained and the results are not as clearly stated as they could be, which lessens the significance of the findings.

Major comments:

- Overall, the context of the study is not well-defined, which makes it hard to evaluate its significance. The Introduction should more fully describe what is already known about how pol theta selects microhomology (MH) regions. The outstanding questions in the field are not referenced. The novel contributions of this *in vitro* study of Pol theta mediated TMEJ should be mentioned.
- An introductory schematic illustrating the TMEJ process and what aspects of Pol theta activity are unknown would be quite useful.
- The first part of the study describing Figs 1-3 was less clearly written. It would help to add a sentence at the beginning of sections setting up what specific attributes are being tested, or at the end of a section summarizing the new knowledge and

significance of the finding.

- Some of the small-scale analysis data (Figs 1B, 1C, 3E, and 3F—which was never described in the text) could be moved to supplemental.
- The way the anchoring position distance was defined in Fig 5B was not clear.
- The reevaluation of previously published data in the last figure seemed inadequate on its own. A more comprehensive analysis of specific mutations using results from multiple studies would greatly strengthen the relevance of the revised definition for a microhomology usable by Pol theta.
- A figure showing the purified Pol θ enzyme on a gel is needed to verify its purity and confirm no other enzymes were present in the in vitro assays.

Minor comments:

- It was hard to follow the layout of several of the figures. For example, in Fig 1, it was not initially clear which data belonged in panel C vs. panel D.
- In the small-scale sequencing (Figs 1-3), the in vitro products were PCR amplified and cloned into plasmids before sequencing. These steps could produce replicate products (for example, by having sister colonies produced from one original transformant in the outgrowth period), so the ratio of different products should not be considered very quantitative.
- In Figure 3B, if both white bands are used for quantitation, were the overlapping green and red signal in the lower white band able to be excluded?
- Certain figure panels were difficult to understand: the schematic in 5B could be expanded to be more clear, and the data analysis in Figures 5A and 6C could be removed or explained more fully in the text.
- In the Methods under TMEJ product measurement: It is not clear what is meant by “the average of the two fluorescence signals was calculated”.

Suggestions to increase manuscript clarity:

- Abstract: After the phrase “Significantly, we found that microhomologies in this region may be interrupted by mismatches”, and “and still used by Pol theta”.
- Introduction: A panel showing the TMEJ mechanism in Fig 1 would be helpful. It would be useful to briefly define “internal microhomologies”.
- Pg. 3, bottom: Three questions are posed here and could be directly referred to throughout the article to draw out the importance of each figure.
- Pg. 4, last sentence: Provide a brief definition of the new conception of microhomology revealed by this article.
- Pg. 7 and 8, bottom: At the end of the section, clearly summarize the new knowledge and significance of the finding.

Version 1:

Reviewer comments:

Reviewer #1

(Remarks to the Author)

The authors have adequately addressed the major concerns regarding the novelty issues raised in the previous manuscript. The revised version includes thoughtful additions to the Discussion and Figures, which clearly articulate how the use of high-throughput sequencing provides novel insights beyond prior biochemical approaches. The explanation of Pol θ 's utilization of extended and mismatched microhomologies is now presented with sufficient clarity through the revised figures and accompanying data and analyses. Overall, the revisions substantially improve the manuscript, and I believe it is suitable for publication in its current form.

(Remarks on code availability)

Reviewer #2

(Remarks to the Author)

The authors have sufficiently addressed our concerns. Specifically, the addition of the experiment with ATP and the relabeling of the specified figures have increased the impact of the study. Furthermore, the reanalysis of the Hussmann et al. TMEJ repair events in the context of the expanded microhomology definition (Figure 8 and Table 1) drives home the point that some of the TMEJ literature will need to be reconsidered in the context of this new paradigm.

The revisions in the text and figures were responsive to the reviewer suggestions and have nicely increased the clarity of the narrative.

Our one final request is that the authors make their code public immediately upon publication of the article, so that others in this field can utilize it to reanalyze their results.

(Remarks on code availability)

We did not install and run the application, but inspection of the README file and code files indicates that the instructions should be sufficient for users who are familiar with Python.

Reviewer #3

(Remarks to the Author)

(Remarks on code availability)

The code appears sufficient and they include a demo dataset to practice with and learn from.

Reviewer #4

(Remarks to the Author)

The changes (textual, graphical, and statistical) have improved the paper, and I have no other concerns.

(Remarks on code availability)

Thank you for the expert reviews and comments on our submitted manuscript. Overall, the reviewers were positive, finding the paper rigorous, comprehensive, and providing new answers for important questions.

The experiments provide answers to the questions now illustrated in the new Fig. 1b: (1) Pol θ usually selects microhomologies interrupted by mismatches (2) The 6 nucleotides closest to the 3' end of a primer are important for determining microhomology choice (3) Bidirectional synthesis is not necessary to initiate end joining.

The first conclusion will likely stimulate the most general interest. This is because the frequent selection of mismatched microhomologies suggests a revision of the definition of microhomology to account for the unique properties of Pol θ . This could advance the analysis of mutations in the genomes of cancers with homologous recombination defects. We now include a full analysis of Pol θ -dependent deletions in a published data set, in a new Figure 8 and Table 1. We found that most of the deletions can be reinterpreted by taking account of mismatched microhomologies. Significantly, this greatly simplifies explanations of how the deletions occurred; we show several examples where deletions thought to have different origins and endpoints are actually caused by the same event. We appreciate the advice of the referees in encouraging this new analysis.

The new Supplementary Fig. 9 shows results of sequencing experiments from TMEJ reaction mixtures where ATP was included. Statistical analysis shows that the distribution of microhomologies is not statistically different from reaction mixtures without ATP. This strengthens the conclusions regarding the intrinsic properties of MH targeting by Pol θ .

As described in more detail below, many aspects of the presentation have been revised and sections rewritten to highlight the novel points, emphasize the context of the findings, and state the conclusions more clearly. Most figures have been revised and reorganized accordingly.

As requested, we provide all source data with this resubmission, and formatting of the text according to *Nature Communications* guidelines.

Reviewer #1 (Remarks to the Author):

In the manuscript, Li and colleagues investigate the mechanisms of microhomology selection by Pol θ during DNA repair. The authors employ a combination of in vitro biochemical assays and high-throughput sequencing analyses to provide new insights into how Pol θ selects and utilizes microhomologies during the end-joining process.

This work is technically sophisticated, offering a single-base resolution analysis of the

microhomology selection mechanism by Pol θ . The data quality is robust and meets the standard for publication in Nature Communications.

However, my main concern lies in the advancement of novelty compared to previous studies. This issue could be addressed by incorporating additional experiments to further strengthen the manuscript before publication.

We appreciate the positive comments on the data quality and importance of our new insights. As mentioned above and in response to this and referee 4, we've now included a new analysis of published data relevant to mismatched microhomologies, in a new Figure 8 (the former Fig. 8 is now Supplementary Fig. 10). We emphasize the importance of the concept of mismatched microhomologies in considering mutations in homologous-recombination defective cancer genomes. We also include new experiments, as suggested, to determine if the inclusion of ATP alters microhomology selection by purified Pol θ .

Major concerns:

This study provides a more comprehensive analysis of Poltheta functions using NGS sequencing compared to previous biochemical studies (e.g., Ref. 9). However, the core findings on mismatch tolerance and minimal base-pairing requirements are similar to those previously reported.

1. To enhance the novelty of the manuscript, the authors should include additional discussions on how the use of NGS data advances the understanding of Poltheta functions beyond conventional biochemical approaches.

This is an important point, and we have now included it in the second paragraph and final page or so of the Discussion section. The high throughput sequencing overcomes the limitations and possible ambiguities associated with traditional approaches for studying Pol θ function. This sequencing analysis reveals, in a way that has not been demonstrated before, that Pol θ uses information from at least 6 nt from the terminus, so that additional pairing is often used that is obscured by the traditional definition of microhomology. The frequent selection of mismatched microhomologies suggests a revision of the definition of microhomology to account for the unique properties of Pol θ , as we discuss in the manuscript and the new Fig. 8 / Table 1. This new view is likely to affect future analysis of mutations in the genomes of cancers with homologous recombination defects.

2. While this study demonstrates that mismatched ends promote internal, imperfectly paired microhomology selection by Poltheta in vitro, the molecular mechanism

underlying this process remains unclear. Additional experimental approaches, such as structural analysis, systematic biochemical assays, and cellular experiments, would be beneficial to elucidate how mismatched ends mechanistically influence internal anchoring by Pol θ .

We fully agree that our findings will encourage further studies of the detailed mechanism of microhomology selection by Pol θ . Structural studies of different protein variants with DNA primers may be particularly informative. No one has yet obtained a structure of Pol θ (or any DNA polymerase) paired at an internal microhomology, but this is a conceivable future area of study. Our study contributes systematic analysis of how Pol θ operates by using primers where a core MH is provided, with random additions of 3' unpaired bases (Figure 4). To compare these outcomes to experiments in cells, Figures 8 and S10 analyze experimental data and provide new insights that will be of general interest to those studying repair. More could certainly be done. In separate experiments, we are analyzing reconstituted TMEJ with Pol θ , Pol δ , RPA, ATP *etc.* We will include purified mutant protein variants in these studies, that show how alterations in protein contacts with DNA can change microhomology selection by Pol θ .

3. Additionally, statistical analysis should be conducted for all quantitative data to ensure the robustness and reliability of the findings.

Statistical analysis has been added to test the significance of distribution differences in Figures 5, 7, S4, S5, and S7, including the new experiments in Figure S9. For experiments performed multiple times, standard errors are indicated in Figure S3b.

Reviewer #2 (Remarks to the Author):

In this work, Li and colleagues leverage an *in vitro* system to characterize the mechanisms by which DNA polymerase theta selects and aligns microhomologies (MHs) during TMEJ. Using a purified form of pol theta with a shortened linker domain, they systematically test libraries of oligonucleotides in end-joining reactions, assessing the importance of microhomology length, imperfect pairing, and non-homologous tails. From their analysis, they draw three important conclusions: (1) pol theta has a strong preference for MHs that are located with 15 nucleotides of the ssDNA ends; (2) pol theta often aligns MHs that contain mismatches, and ssDNA with multiple, non-contiguous mismatches can be substrates for joining; and (3) pol theta can utilize its polymerase activity to extend one annealed strand independently of the other.

Many cellular and organismal studies have shown that pol theta prefers to join MHs that are located close to the 3' ends of breaks; conclusion 1 validates these. Others have also suggested that MMEJ/TMEJ might involve alignment of MHs with mismatches, but

to my knowledge this is the first rigorous experimental demonstration of this. The same is true for the third conclusion listed above, and the authors may want to stress this novelty. Taken together, the findings from this study highlight the unusual flexibility of TMEJ and further expand the range of end-joining repair events that might be attributed to this error-prone repair process. If the findings are generalizable to more physiologically relevant contexts (see below), many studies that have assigned double-strand break repair products to either NHEJ, MMEJ, or TMEJ might need to be reinterpreted.

We appreciate these comments – and we have emphasized the conclusion points in the Abstract, Introduction and Discussion sections.

However, there are some weaknesses that need to be addressed to increase confidence that these data could represent the actual TMEJ mechanism in a cellular context.

First, although the authors are using purified pol theta containing both helicase and polymerase domains, they don't include ATP in the end-joining reactions. It is possible that the helicase/ATPase domain of pol theta could unwind stem-loop extension products or disrupt poorly annealed MHs in an ATP-dependent manner. In addition, as the authors mention in the discussion, ssDNA is usually bound by RPA *in vivo* and needs to be stripped via pol theta helicase activity to promote efficient MH annealing. To investigate these possibilities, experiments that are critical to the conclusions in figures 1-3 should be repeated with ATP (at a minimum) and purified RPA if possible.

We agree that many further investigations along these lines are possible. The focus of the current report paper is to understand the capabilities of Pol θ – by itself – to find MH anchoring positions. Because it is so relevant *in vivo*, in this revised version we have added experiments as suggested to compare previous results without ATP to those with ATP. The data with ATP are included in the new Supplementary Fig. 9. For this experiment, we repeated the experiment with all substrates that were in Fig. 3b and obtained 700 new DNA sequences. Statistical analysis shows that the distribution of MH anchoring patterns is not different in conditions with and without ATP (Supplementary Fig. 9b & c). This is described on page 8 of the revision. The result is perhaps not unexpected; both experiments include dNTPs, and Pol θ can hydrolyze dATP as well as ATP. Regarding whether Pol θ can help unwind stem-loop extension products, a previous study showed that these remain stable in conditions where Pol θ is hydrolyzing dNTPs (Figure 7 of Carvajal, Li *et al* JBC 2024 PMID: 38876299). The new Supplementary Fig. 9 shows that the proportion of stable stem-loop products remains the same with and without ATP in the reaction mixture. These points are now included in

the results and discussion sections. Regarding how RPA modulates anchoring positions, we are including this experiment in a manuscript in preparation where we describe reconstitution of TMEJ with Pol θ , Pol δ , and RPA. The results so far show that the characteristics of the anchoring MH are similar, with many including mismatches, and that anchoring positions are shifted more towards the 3' end of the DNA tail.

Second, in many of the reactions the use of internal, imperfectly paired MHs to promote end joining is necessitated due to the presence of non-homologous 3' tails. The Ramsden lab has shown that these tails can be removed through exonuclease activity of pol delta. If this occurs, and MH annealing is transient, it seems possible that some of the mismatched MHs might be disrupted by either pol delta exonuclease action or pol theta helicase activity, leading to a different spectrum of end-joining products than what was observed in these studies. Ideally, this should be tested using purified pol delta, but perhaps the exonuclease activity of pol delta is so robust that the ssDNA gets rapidly and non-specifically cleaved in vitro (Fig. S4)? Regardless, this caveat needs to be included in the discussion.

Regarding the involvement of Pol δ , we have now emphasized this point more in the Discussion. The mismatched microhomology selection preference that we describe is directly relevant to combined action of the two enzymes. After Pol δ acts on a 3' end to trim a base, this creates a new end for a new microhomology search. We are working on a large set of experiments that we plan to include in a future report as indicated above. So far, those results show that Pol δ trims off one nucleotide at a time from a 3' end and then passes the 3' tail back to Pol θ , where it initiates a new microhomology search.

Other minor points:

1. The distribution of MH anchoring positions in Fig. 3f doesn't match the conclusion drawn from the data in Fig. 5. Is this because the MH anchoring position is being defined differently in Fig. 3?

This is an important comment as it emphasizes that we needed to be clearer about explaining the differences between the experiments and their respective purposes. Figure 3 is presented as an introductory experiment, using only seven oligonucleotides and small-scale sequencing. Either the top or the bottom strand can extend (the bottom strand is unblocked "oligo 7"). When the top strand has unpaired bases at the 3' end, the bottom strand is extended more efficiently and prefers anchoring at three specific sites (e.g., Fig 3d). We moved the former Fig 3f to Supplementary Fig. 3e. This shows the summary results for the six oligonucleotides with unpaired bases at the 3' end, and it is true that for this special case, many pairings occur further than 15-20 nt from the terminus.

To understand the more general case, we turned to high-throughput sequencing for the experiments in Fig. 5. Here, the bottom strand is blocked (“oligo 10”) so that extension of top strands can be unambiguously assigned. Looking at the this much larger combined data set (1024 extended oligonucleotides in the final set), the data show that most – but not all – anchoring takes place within 15 nt of the terminus.

Similarly, why do the anchoring positions in Fig. 5 only extend to 40 nt to the right/left of the 3' ends?

From Fig. 4b, it appears that there are 68 nt to the left of the ssDNA ends that are being sequenced. This needs to be clarified.

We've revised the labeling in Fig. 4b and the Fig. 4b legend to clarify all this. Figure 5 uses the blocked bottom oligonucleotide, so that extension only takes place from the top strand to the right, for 40 nt. The PCR amplification primer is designed to anneal with a priming start at the position adjacent to this 40 nt.

In Fig. 7, we show data from the high throughput experiments with unblocked bottom strand (“oligo 8”). Here, only anchoring at positions up to 35 nt to the left can be analyzed, because the oligonucleotide barcode is adjacent to that position.

2. Fig. 6c: We had difficulty understanding how this clustering diagram demonstrates that matched base pairs are not usually consecutive. If they were consecutive, would each column be expected to contain an unbroken blue bar that spans at least part of the P1-P6 space? A more complete explanation here would help.

Yes, in Fig. 6c, each column corresponds to the most-favored MH for a specific individual oligonucleotide in Library 6-10. For each column the blue color is uniform as it indicates the proportion of this outcome amongst all outcomes for each oligonucleotide. If a position is white, it indicates a mismatch. Thus, the great majority of favored MH are interrupted by mismatches. To make this clear, so we have expanded and edited the Figure 6c legend and the relevant text.

3. For consistency, the percentage of TMEJ products should be shown below the gels in Fig. 2A&C.

Good point – this is now done.

4. Figure 2 legend title: The use of the word ‘terminal’ is somewhat misleading here, since oligo 2N2M has a 2-base pair mismatched 3' terminus. We suggest using ‘core MH,’ as was used at the bottom of page 7.

Also an excellent point – this is now changed to “core MH”.

Reviewer #3 (Remarks to the Author):

Reviewer #4 (Remarks to the Author):

DNA polymerase theta contributes to double-strand break (DSB) repair by extending single stranded DNA ends imperfectly paired at regions of limited homology. This study characterized Pol theta sequence preferences using an in vitro biochemical system coupled with high-throughput sequencing. The authors demonstrated that Pol theta can extend from terminal mismatches and then characterized its patterns of sequence usage. They observed that a match at the 3'-most nucleotide contributes the most to microhomology site selection, that end-joining occurs more frequently at regions with larger numbers of matched basepairs (up to six), and that mismatches within the six nucleotide microhomology region are allowed. Using their findings, the authors reinterpreted published data from an in vivo DSB repair assay done in human cells.

The article addresses the important mechanistic question of how Pol theta selects regions of microhomology, which are only a few bp in length. However, the context of the study is not fully explained and the results are not as clearly stated as they could be, which lessens the significance of the findings.

Major comments:

- Overall, the context of the study is not well-defined, which makes it hard to evaluate its significance. The Introduction should more fully describe what is already known about how pol theta selects microhomology (MH) regions. The outstanding questions in the field are not referenced. The novel contributions of this in vitro study of Pol theta mediated TMEJ should be mentioned.

Thank you for the careful review, particularly the many comments that have increased the clarity of the presentation. This is a valuable overall comment because it shows that we had to better describe how this study fits into current overall knowledge, the outstanding contributions, and briefly presage the outstanding contributions (this is also true for the abstract).

- An introductory schematic illustrating the TMEJ process and what aspects of Pol theta activity are unknown would be quite useful.

This is useful advice. Fig 1 a and b have been added to the revision. In the Introduction, there is now a better exposition about Pol θ and TMEJ, and questions to be answered.

- The first part of the study describing Figs 1-3 was less clearly written. It would help to add a sentence at the beginning of sections setting up what specific attributes are being tested, or at the end of a section summarizing the new knowledge and significance of the finding.

Thanks – this has been done for Figs 1-3, and elsewhere. We think the flow and explanation is much better.

- Some of the small-scale analysis data (Figs 1B, 1C, 3E, and 3F—which was never described in the text) could be moved to supplemental.

If possible, we prefer to include some of these diagrams in the main text because they set the stage for the analysis and graphically show that internal MH are used, which justifies the further analysis. We have improved the discussion of each in the manuscript. As recommended, the former Fig 1b has been moved to Supplementary Fig 1b and Fig 3e-f have been relocated to Supplementary Fig 3d & e.

- The way the anchoring position distance was defined in Fig 5B was not clear.

We modified the figure with labels to explain this more clearly and adjusted the text and figure legend.

- The reevaluation of previously published data in the last figure seemed inadequate on its own. A more comprehensive analysis of specific mutations using results from multiple studies would greatly strengthen the relevance of the revised definition for a microhomology usable by Pol theta.

It's a good suggestion; to do this properly we needed data from a study that clearly separated Pol θ dependent and Pol θ independent events. We found such a published study that examined many DSB sites (Hussmann *et al.* 2021, *Cell* 184: 5653-5669 e25). All of the deletions that were strongly Pol θ -dependent were determined and reanalyzed. This data appears in the new Fig 8, with the previous Fig 8 moved to Supplementary Fig. 10. The conclusion is actually more remarkable than we expected. Not only could most events be reinterpreted as arising from mismatched microhomologies, but multiple events could also be explained as having the same

original pairing. This increases our confidence that this finding is going to generate interest in the field. In the future it would be possible to evaluate, for example, deletions in BRCA-deficient cancers to determine predicted mismatched microhomologies. But in these cases, we do not have firm evidence of Pol θ dependence, and it will be best to do this aligned with other systematic screens.

- A figure showing the purified Pol ϕ Δ cen enzyme on a gel is needed to verify its purity and confirm no other enzymes were present in the in vitro assays.

This gel image for purified Pol θ has been added as Fig 1a. We detect no major contaminants; Supplementary Fig 4d shows no observed nuclease contamination.

Minor comments:

- It was hard to follow the layout of several of the figures. For example, in Fig 1, it was not initially clear which data belonged in panel C vs. panel D.

This is important. We changed the layout and introduced more space to separate the panels in Fig 1 and other figures.

- In the small-scale sequencing (Figs 1-3), the in vitro products were PCR amplified and cloned into plasmids before sequencing. These steps could produce replicate products (for example, by having sister colonies produced from one original transformant in the outgrowth period), so the ratio of different products should not be considered very quantitative.

Yes, the small-scale sequencing represents relative abundance, but it serves to show the type and range of possible outcomes. This motivated us to design a high-throughput sequencing approach with an individual barcode identifier for each product.

- In Figure 3B, if both white bands are used for quantitation, were the overlapping green and red signal in the lower white band able to be excluded?

Yes, only the TMEJ products (now more clearly marked in the figure) are measured. The measurement was done separately for green (Cy3 channel) and magenta (Cy5 channel) signal via ImageJ. The individual stem-loop products tend to migrate together on the gel in reactions with Pol θ Δ Cen, but these are well separated from the TMEJ products and not used for quantification.

- Certain figure panels were difficult to understand: the schematic in 5B could be

expanded to be more clear, and the data analysis in Figures 5A and 6C could be removed or explained more fully in the text.

The labeling in Fig 5b has been expanded with a better explanation in the figure legend and accompanying text. We hope to keep Fig 5a and 6c in the main text. Fig 5A shows that with Pol θ alone, the major outcomes were deletion events rather than insertions and complex events. We have also worked on improving the presentation and explanation of Fig 6C in the legend, as described above for Reviewer #2. It illustrates that preferred microhomologies usually contain mismatched bases, an important conclusion.

- In the Methods under TMEJ product measurement: It is not clear what is meant by “the average of the two fluorescence signals was calculated”.

The procedure is explained in the Methods, page 20.

Suggestions to increase manuscript clarity:

- Abstract: After the phrase “Significantly, we found that microhomologies in this region may be interrupted by mismatches”, and “and still used by Pol theta”.

We reworked the Abstract to make it clearer, we hope, it now includes the phrase “... the microhomologies selected for extension are usually interrupted by mismatches”

- Introduction: A panel showing the TMEJ mechanism in Fig 1 would be helpful. It would be useful to briefly define “internal microhomologies”.

- Pg. 3, bottom: Three questions are posed here and could be directly referred to throughout the article to draw out the importance of each figure.

Yes, we’ve added Fig. 1b to take care of these points and outlined this in the text.

- Pg. 4, last sentence: Provide a brief definition of the new conception of microhomology revealed by this article.

Thanks, we now end the Introduction with a better statement of this concept.

- Pg. 7 and 8, bottom: At the end of the section, clearly summarize the new knowledge and significance of the finding.

Yes, conclusions at the end of this section are now better emphasized.

Thank you for all your expert reviews and comments on our resubmitted manuscript. As requested, we provide all source data with this resubmission, formatting of the text according to *Nature Communications* guidelines, and releasing the raw dataset in SRA and code in Code Ocean.

Reviewer #1 (Remarks to the Author):

The authors have adequately addressed the major concerns regarding the novelty issues raised in the previous manuscript. The revised version includes thoughtful additions to the Discussion and Figures, which clearly articulate how the use of high-throughput sequencing provides novel insights beyond prior biochemical approaches. The explanation of Pol θ 's utilization of extended and mismatched microhomologies is now presented with sufficient clarity through the revised figures and accompanying data and analyses. Overall, the revisions substantially improve the manuscript, and I believe it is suitable for publication in its current form.

We appreciate your careful review and positive comments.

Reviewer #2 (Remarks to the Author):

The authors have sufficiently addressed our concerns. Specifically, the addition of the experiment with ATP and the relabeling of the specified figures have increased the impact of the study. Furthermore, the reanalysis of the Hussmann et al. TMEJ repair events in the context of the expanded microhomology definition (Figure 8 and Table 1) drives home the point that some of the TMEJ literature will need to be reconsidered in the context of this new paradigm.

The revisions in the text and figures were responsive to the reviewer suggestions and have nicely increased the clarity of the narrative.

Our one final request is that the authors make their code public immediately upon publication of the article, so that others in this field can utilize it to reanalyze their results.

We appreciate your positive comments on ATP experiment and reanalysis data. We will release the code and raw data upon publication.

Reviewer #2 (Remarks on code availability):

We did not install and run the application, but inspection of the README file and code files indicates that the instructions should be sufficient for users who are familiar with Python.

The demo code works well in Code Ocean and we believe it's good basic start for analysis.

Reviewer #3 (Remarks to the Author):

Reviewer #3 (Remarks on code availability):

The code appears sufficient and they include a demo dataset to practice with and learn from.

We appreciate your careful review and comments.

Reviewer #4 (Remarks to the Author):

The changes (textual, graphical, and statistical) have improved the paper, and I have no other concerns.

We appreciate your careful review and comments.